# GJA1 depletion causes ciliary defects by affecting Rab11 trafficking to the ciliary base

**Dong Gil Jang[1], Keun Yeong Kwon[1], Yeong Cheon Kweon[1], Byung-gyu Kim[2], Kyungjae Myung[2], Hyun-Shik Lee[3], Chan Young Park[1], Taejoon Kwon[2,4]\*, Tae Joo Park[1,2]\***

[1]Department of Biological Sciences, Ulsan National Institute of Science and Technology, Ulsan, Republic of Korea; [2]Center for Genomic Integrity, Institute for Basic Science, Ulsan, Republic of Korea; [3]KNU-Center for Nonlinear Dynamics, CMRI, School of Life Sciences, BK21 Plus KNU Creative Bio Research Group, College of Natural Sciences, Kyungpook National University, Daegu, Republic of Korea; [4]Department of Biomedical Engineering, Ulsan National Institute of Science and Technology, Ulsan, Republic of Korea

**Abstract** The gap junction complex functions as a transport channel across the membrane. Among gap junction subunits, gap junction protein α1 (GJA1) is the most commonly expressed subunit. A recent study showed that GJA1 is necessary for the maintenance of motile cilia; however, the molecular mechanism and function of GJA1 in ciliogenesis remain unknown. Here, we examined the functions of GJA1 during ciliogenesis in human retinal pigment epithelium-1 and *Xenopus laevis* embryonic multiciliated-cells. GJA1 localizes to the motile ciliary axonemes or pericentriolar regions beneath the primary cilium. GJA1 depletion caused malformation of both the primary cilium and motile cilia. Further study revealed that GJA1 depletion affected several ciliary proteins such as BBS4, CP110, and Rab11 in the pericentriolar region and basal body. Interestingly, CP110 removal from the mother centriole was significantly reduced by GJA1 depletion. Importantly, Rab11, a key regulator during ciliogenesis, was immunoprecipitated with GJA1 and GJA1 knockdown caused the mislocalization of Rab11. These findings suggest that GJA1 regulates ciliogenesis by interacting with the Rab11-Rab8 ciliary trafficking pathway.

**\*For correspondence:**
tkwon@unist.ac.kr (TK);
parktj@unist.ac.kr (TJP)

**Competing interest:** The authors declare that no competing interests exist.

## Editor's evaluation

This important paper solidly demonstrates that the gap junction protein GJA1 localizes to motile cilia and is required for the formation of motile cilia on the developing frog epidermis. In addition, GJA1 localizes to the peri-centrosomal region of primary cilia where it is proposed to participate in Rab8-Rab11 delivery of ciliary cargo. These findings point to new functions for gap junction proteins and raise important questions about the role of Gja1 in ciliary assembly and function.

## Introduction

A gap junction, which is also known as a connexon, is a transmembrane (TM) protein complex that plays an important role as a channel and transports low-molecular-weight compounds, nutrients, and ions laterally across the plasma membrane (*Goodenough et al., 1996*; *Goodenough and Paul, 2009*). It also regulates cell proliferation, differentiation, growth, and death (*Aasen, 2015*; *El-Sabban et al., 2003*; *Vinken et al., 2006*). One gap junction channel (Connexon) is composed of twelve gap junction

protein subunits (Connexin) (*Figure 1A*). And there are 21 human gap junction protein subunit families, which have been classified as five subgroups according to their amino acid sequence homology (*Bosco et al., 2011*; *Söhl and Willecke, 2004*).

Gap junction protein α1 (GJA1), also known as connexin 43 (CX43, 43 kDa), was originally identified in the rat heart (*Beyer et al., 1987*). The human homolog of *GJA1* is located at human chromosome 6q22-q23. The *GJA1* gene was identified after discovering the putative genes for human oculodentodigital dysplasia (ODDD; *Boyadjiev et al., 1999*; *Paznekas et al., 2003*). Among all gap junction protein families, GJA1 is the most common and is a major subunit. It is expressed in many tissues, such as the eyes, ears, brain, and especially the heart (*Oyamada et al., 2005*; *Evans and Martin, 2002*; *Ruangvoravat and Lo, 1992*; *Willecke et al., 2002*).

The GJA1 protein has four TM domains with five linked subdomain loops. Among these loops, two loops are extracellular subdomains, and three loops, including the N- and C-terminal domains, are in the cytoplasm (*Figure 1A*; *Beyer et al., 1990*). In the cytoplasm, the C-terminal domain of GJA1 regulates the cytoskeletal network (actin and tubulin [*Leithe et al., 2018*]) and cell extension, migration, and polarity (*Matsuuchi and Naus, 2013*; *Kameritsch et al., 2012*; *Rhee et al., 2009*). A recent study reported that GJA1 also regulates the maintenance of cilia (*Zhang et al., 2020*), which are microtubule-based cellular organelles that play crucial roles in the physiological maintenance of the human body (*Gerdes et al., 2009*). However, the molecular mechanisms and roles of GJA1 in the formation and function of cilia have not been determined.

Cilia exist in most types of vertebrate cells and are involved in various developmental processes and physiological responses from embryos to adults. For example, cilia are involved in cell cycle control, cell-to-cell signal transduction, fertilization, early embryonic development, extracellular environment sensing, and homeostasis (*Malicki and Johnson, 2017*). Mutations in essential genes for cilia formation disrupt ciliary structures or their functions and lead to 'ciliopathy', which is an innate genetic and syndromic disorder (*Waters and Beales, 2011*; *Fliegauf et al., 2007*).

In this report, we show that GJA1 regulates ciliogenesis by affecting ciliary trafficking and thereby promoting uncapping of the mother centriole. GJA1 is localized not only at the gap junction but also in the pericentriolar regions beneath the primary cilium of retinal pigment epithelium-1 (RPE1) cells and the ciliary axonemes of multiple/motile cilia.

Using immunoprecipitation-mass spectrometry (IP-MS) analysis, we identified Rab11 and Rab8a as putative GJA1-binding partners. Rab11 was previously shown to accumulate around the basal body and is involved in the early steps of ciliogenesis (*Knödler et al., 2010*; *Westlake et al., 2011*). Further analysis showed that Rab11 immunoprecipitated with GJA1, and knockdown of GJA1 in RPE1 cells resulted in the mislocalization of Rab11. These data suggest that GJA1 contributes to proper ciliogenesis by regulating Rab11 trafficking to basal bodies and facilitating ciliary axoneme formation and assembly.

## Results

### GJA1 is localized at the ciliary axonemes and near the basal bodies in multiciliated cells

GJA1, which is a major component of the gap junction complex, consists of four TM domains (TM1–4), interdomain loops between each TM domain, and intracellular N- and C-terminus domains (*Figure 1A*). To investigate the biological effects of GJA1 on ciliogenesis in vertebrates, we cloned the *Xenopus laevis* homolog of human *GJA1* based on a sequence from the Xenbase database (*James-Zorn et al., 2013*). Next, we analyzed GJA1 localization in multiciliated cells in *Xenopus* embryos expressing GJA1-Flag or HA-tagged protein by immunofluorescence staining. We observed that GJA1-Flag strongly accumulated at the cell-cell junction as a gap junction complex (*Figure 1B*, white arrowhead). Unexpectedly, we also observed that GJA1 was localized near the apical surface as puncta (*Figure 1B and B'*, yellow arrow) and in ciliary axonemes (*Figure 1C and C'*, white arrow), in addition to gap junctions in *Xenopus* multiciliated epithelial cells. We further analyzed apical localization of GJA1 puncta by co-immunostaining with a basal body marker, centrin. Interestingly, GJA1-HA signals did not overlap with centrin signals. Instead, GJA1-HA was localized below or above the basal bodies (*Figure 1D and D'*, *Figure 1—figure supplement 1A*). Additionally, we examined the endogenous GJA1 location in multiciliated cells in the mouse branchial epithelium and found that endogenous GJA1 localizes to the

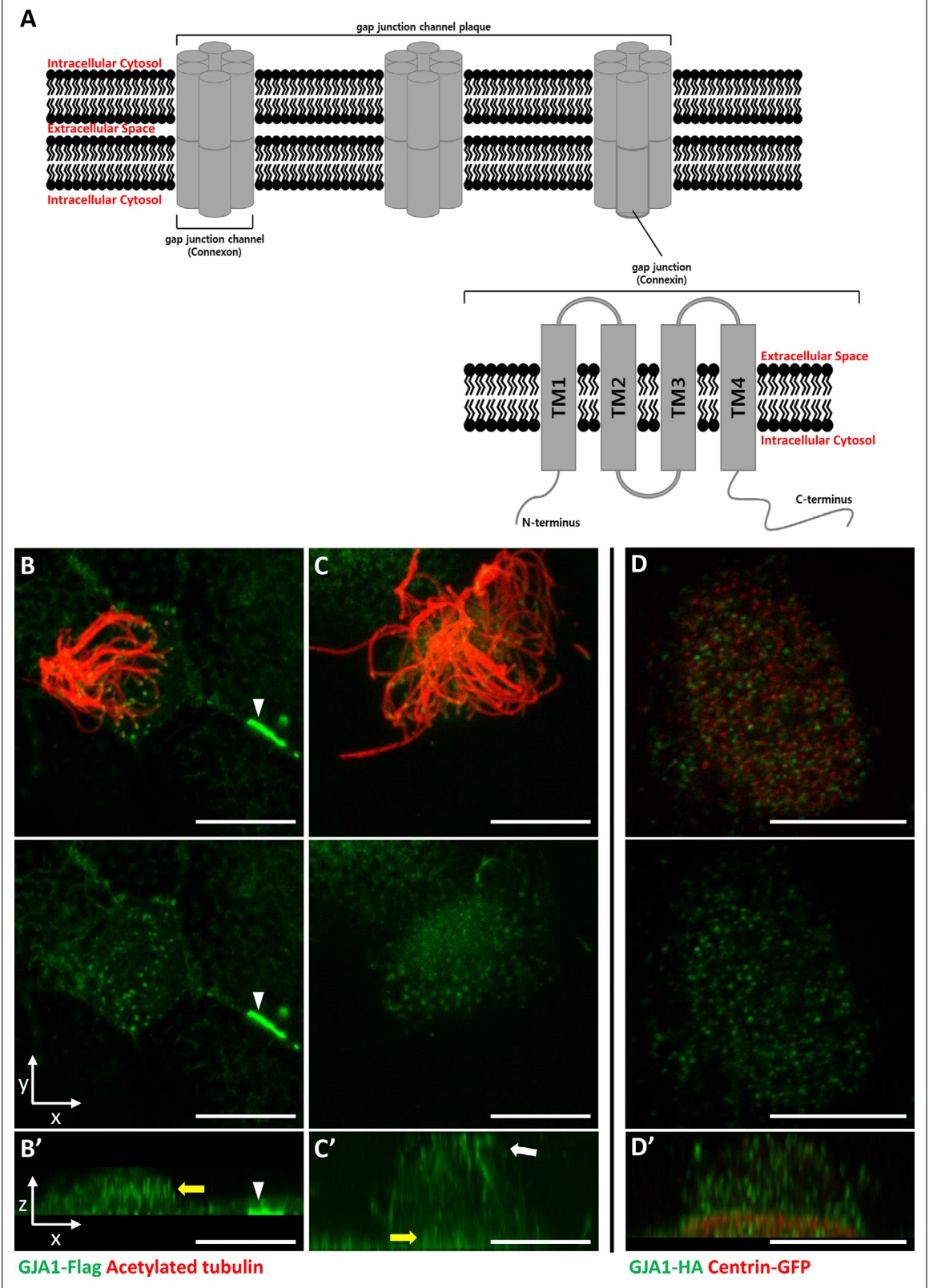

**Figure 1.** GJA1 is localized at the ciliary base and axonemes in addition to gap junctions in *Xenopus* multiciliated epithelial cells. (**A**) The schematic structure of gap junction channels and GJA1. (**B, C**) Immunofluorescence analysis of the localization of GJA1 in *Xenopus* multiciliated epithelial cells. *Xenopus* embryos were microinjected with *GJA1*-Flag mRNA, and the embryos were stained with antibodies against acetylated tubulin (red) and flag-tag (green). Scale bars: 10 μm. (**D**) Immunofluorescence analysis of the localization of GJA1 and the basal body marker centrin in *Xenopus* multiciliated

*Figure 1 continued on next page*

*Figure 1 continued*

epithelial cells. *Xenopus* embryos were co-microinjected with *GJA1*-HA mRNA and *centrin*-GFP mRNA. The embryos were stained with antibodies against HA-tag (green) and GFP (red). Scale bars: 10 μm B′–D′. X-Z projection images of panels (**B–D**).

The online version of this article includes the following figure supplement(s) for figure 1:

**Figure supplement 1.** GJA1 is localized to ciliary axonemes in *Xenopus* multiciliated cells and mouse tracheal tissues.

ciliary axonemes as puncta (*Figure 1—figure supplement 1B*). These data suggest that GJA1 may be involved in ciliogenesis.

## Dominant-negative mutant-mediated dysfunction of GJA1 causes severe ciliary malformation in *Xenopus* embryonic epithelial tissue

To determine the function of GJA1 during cilia formation, we exploited dominant-negative mutants of GJA1 (dnGJA1). Two known dominant-negative mutants have been identified and were used in recent studies. The T154A point mutation mimics the closed-channel status of the gap junction complex but does not inhibit gap junction formation (*Beahm et al., 2006*). The Δ130–136 deletion mutation consists of a seven-amino acid deletion in the intracellular loop between TM2 and 3 that blocks gap junction permeability (*Wang et al., 2005*). Last, we designed the Δ234–243 mutant, which consists of a 10-amino acid deletion in the putative tubulin-binding sequence of the C-terminus. This sequence only exists in GJA1 and is not conserved in other gap junction protein families (*Figure 2—figure supplement 1A*; *Giepmans et al., 2001*). Microinjection of each dominant-negative mutant at ventral-animal regions of two-cell stage embryos caused severe defects in cilia formation (*Figure 2A*). Immunofluorescence analysis using an acetylated tubulin antibody showed severely shortened and fewer ciliary axonemes in the dnGJA1-injected embryos compared with those of control embryos (*Figure 2A*). In particular, the number of cilia on ciliated cells was notably decreased in Δ234–243 mutant-injected embryos, although it would be challenging to quantitatively analyze the phenotypes (*Figure 2A*). However, the basal bodies, which are the organizing center of ciliary axonemes, did not have noticeable defects, and the number of basal bodies and their apical localization were comparable to those of controls (*Figure 2A*, green). Next, we measured the number of ciliated cells by counting acetylated tubulin-positive cells, and indeed the number of ciliated cells was significantly reduced in the dnGJA1-expressing embryos (*Figure 2B*). We next isolated cilia from wild-type embryos or dnGJA1-injected embryos and analyzed the average length of isolated cilia. The ciliary length was significantly shorter in the dnGJA1 expressing embryos (*Figure 2—figure supplement 1B,C*).

Then, we examined localization of each dnGJA1 in multiciliated cells by expressing dn*GJA1*-Flag mRNAs and observed that the Δ234–243 mutant did not exhibit ciliary axoneme localization, and the Δ130–136 mutant displayed reduced axonemal localization (*Figure 2—figure supplement 1D*). From these observations, we speculated that dnGJA1 may interfere with the proper localization of wild-type GJA1(wtGJA1) and thereby disrupt cilia formation. To test this idea, we co-injected HA-tagged wild-type GJA1 (wt*GJA1*-HA) and dn*GJA1*-Flag mRNA and analyzed the changes in wtGJA1-HA localization in multiciliated cells. As we speculated, co-injection of dnGJA1-Flag with wtGJA1-HA resulted in displacement of wtGJA1-HA from the ciliary axonemes (*Figure 2C and D*). These data suggest that the GJA1 function is necessary for proper cilia formation, especially ciliary axoneme assembly.

## Morpholino-mediated knockdown of *GJA1* disrupts normal ciliogenesis in *Xenopus*

Next, we sought to further confirm the necessity of GJA1 in ciliogenesis by depleting proteins using an antisense morpholino oligo (*GJA1*-MO). To determine whether antisense MO-mediated knockdown of *GJA1* also causes similar phenotypes to those of dominant-negative mutants, we designed an antisense MO to block translation by binding to the transcription start site. Antisense MO injection effectively reduced the translation of wild-type *GJA1*-Flag mRNA (*Figure 3A*). In contrast, the translation of mismatch *GJA1* mRNA (mis-m*GJA1*), which was modified to be mismatched to the MO sequence, was not affected as strongly as that of wild-type mRNA (*Figure 3A*, MO + mis mRNA). These data indicate that *GJA1*-MO efficiently inhibits wild-type *GJA1* mRNA translation. To test for specificity, we further performed a rescue experiment (see below).

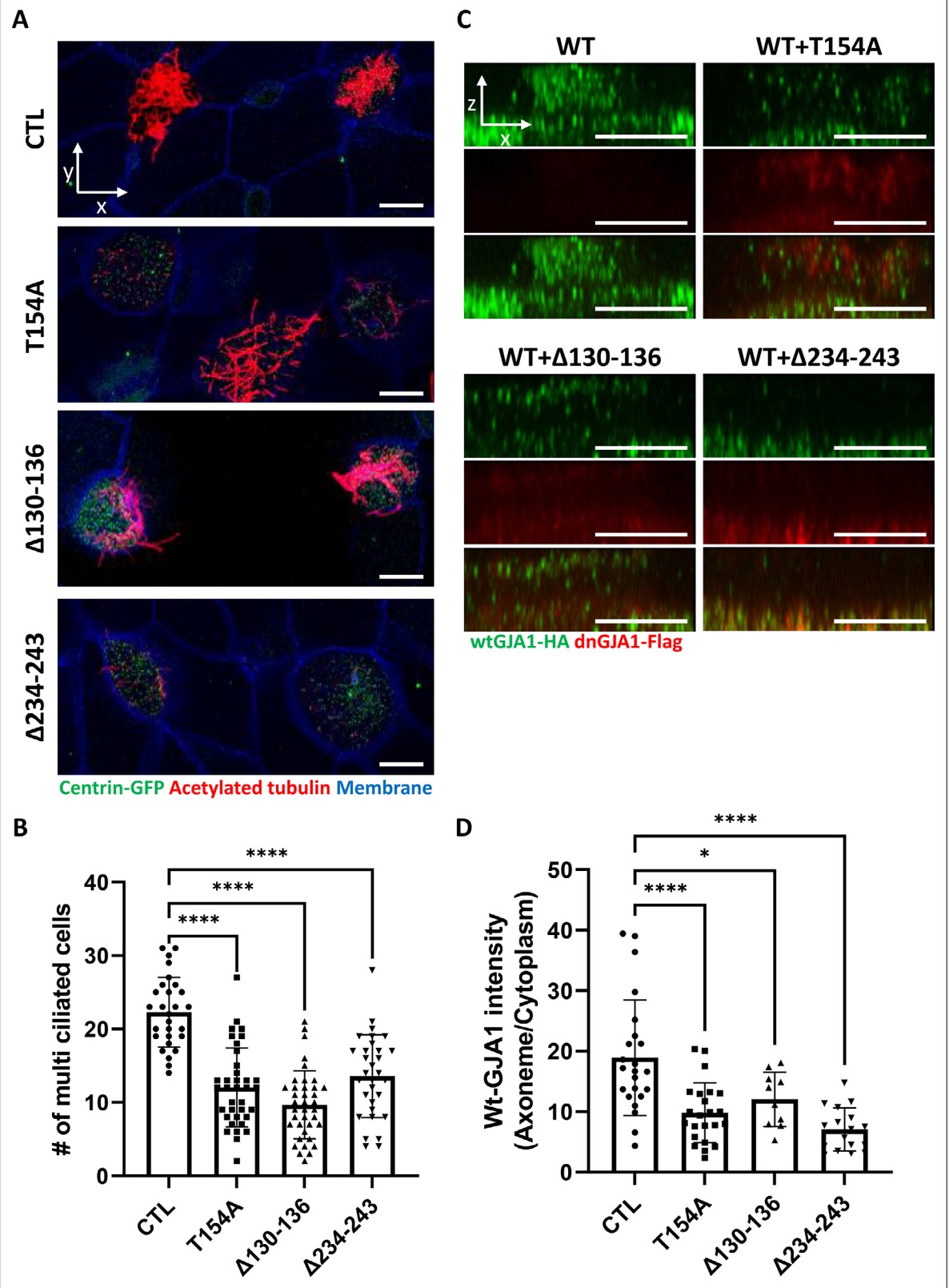

**Figure 2.** GJA1 dysfunction causes ciliary defects in *Xenopus* multiciliated epithelial cells. (**A**) *Xenopus* embryos were injected with the indicated dn*GJA1* mRNAs, and the mucociliary epithelium was immunostained. The ciliary axonemes were stained with acetylated tubulin antibody (red). Basal bodies were visualized by microinjection of *centrin*-GFP (green). Plasma membranes were labeled with membrane RFP and detected using an anti-RFP antibody (blue). Scale bars: 10 μm. (**B**) Statistical analysis of the number of multiciliated cells per unit area in panel (**A**). Error bars represent the mean ±

*Figure 2 continued on next page*

*Figure 2 continued*

SD. P values were determined by one-way ANOVA (p****<0.0001). n=28(CTL), 36(T154A), 38(Δ130-136), 31(Δ234-243). Raw values are provided in the *Figure 2—source data 1* file. (**C**) *Xenopus laevis* embryos were co-microinjected with HA-tagged wild-type (WT) *GJA1* (wt*GJA1*-HA) and flag-tagged dominant-negative mutant *GJA1* (dn*GJA1*-Flag) mRNAs. Wild-type GJA1 was stained with an HA antibody (green), and each dominant-negative mutant GJA1 was stained with a flag antibody (red). Scale bars: 10 μm. (**D**) Statistical analysis of GJA1 intensity as a ratio of that in the ciliary axoneme to that in the cytoplasm in panel (**C**). Error bars represent the mean ± SD. P values were determined by one-way ANOVA (p****<0.0001, p*=0.0194). n=23(CTL,T154A), 10(Δ130-136), 17(Δ234-243). Raw values are provided in the *Figure 2—source data 2* file.

The online version of this article includes the following source data and figure supplement(s) for figure 2:

**Source data 1.** The number of multiciliated cells per unit area.

**Source data 2.** The intensity of GJA1 signal as a ratio of that in the ciliary axoneme to that in the cytoplasm.

**Figure supplement 1.** Dominant-negative GJA1 mutants cause ciliary defects and exhibit differential localization.

**Figure supplement 1—source data 1.** The average lengths of ciliary axonemes.

We next analyzed the phenotype of *GJA1*-MO-injected embryos. Consistent with the mutant-based approaches, multiciliated cells in *GJA1*-MO-injected embryos (morphants) were severely malformed compared to those of control embryos (*Figure 3B*), but basal bodies were not severely affected by *GJA1*-MO injection (*Figure 3B*, green). Moreover, MO-mediated knockdown of *GJA1* resulted in a reduction of acetylated tubulin-positive multiciliated cells in *Xenopus* embryonic epithelial tissues (*Figure 3C* middle panel, E). Co-injection of MO-mismatched-*GJA1* mRNA (mis-m*GJA1*) with *GJA1*-MO effectively rescued the ciliary defects (*Figure 3C* right panel, D, E), indicating that defects in cilia formation in GJA1 morphants are a specific loss-of-function phenotype of GJA1. We also co-injected each dn*GJA1* mRNA with *GJA1*-MO. However, dominant-negative mutants failed to exhibit reversal of the ciliary defects seen in the GJA1 morphants (*Figure 3D and E*).

Based on the loss-of-function phenotype of GJA1 morphant embryos, we further determined whether GJA1 may affect cell fate specification during ciliated cell differentiation or whether it mainly affects ciliogenesis after cell fate determination. To this end, we performed whole-mount in situ hybridization using a *DNAH9* antisense probe, which is a component of the ciliary outer dynein arm and is a known multiciliated cell marker (*Fassad et al., 2018*). In contrast to the reduction of acetylated tubulin-stained multiciliated cells in the embryonic epidermis of *Xenopus*, the number of *DNAH9*-positive multiciliated cells in GJA1 morphants was not significantly different from that in controls (*Figure 3—figure supplement 1A*, B), which is implicating GJA1 did not affect the cell fate determination.

We further confirmed the GJA1 loss-of-function phenotype using CRISPR/Cas9-mediated $F_0$ mutagenesis by micro-injecting Cas9 and guide RNA (*GJA1*-Cas9) into two-cell stage embryos. Sequencing analysis and an in in vitro Cas9 reaction using a PCR product covering the target site revealed that *GJA1*-Cas9 injection effectively induced indel mutations at the *GJA1*-Cas9 target sites (*Figure 3—figure supplement 2A,B*). Consistent with the GJA1 morphant phenotypes, we observed a reduction of acetylated tubulin-positive multiciliated cells (*Figure 3—figure supplement 2C,D*). These data indicate that GJA1 has previously unidentified but critical functions in cilia formation.

## GJA1 function is necessary for gastrocoel roof plate cilium formation and left-right asymmetry development

Motile cilia are not only involved in mucus clearance in the mucociliary epithelium but are also critical for determining left-right asymmetry during early embryonic development. Embryonic nodal cilium generates leftward flow, and this flow initiates the left lateral plate mesoderm (LPM) specification. In *Xenopus*, the gastrocoel roof plate (GRP) possesses a nodal cilium, and this mono-, motile nodal cilium is necessary for formation of left-right asymmetry (*Schweickert et al., 2007*; *Blum et al., 2009*). Therefore, we next determined whether GJA1 is also necessary for GRP cilium formation. *GJA1*-MO was injected into the two dorsal-vegetal cells in four-cell stage embryos to target GRP tissues. Indeed, GJA1 morphants displayed abnormally shortened nodal cilium in the GRP compared to those of control embryos (*Figure 4A and B*). Next, we assessed left-right asymmetry by observing heart development in later embryonic stages. In the ventral view, the truncus arteriosus of the control *Xenopus* embryonic heart was placed to the left of the embryos by looping of the primitive heart tube. We observed that approximately 30% of morphant embryos developed situs inversus (reversed organs;

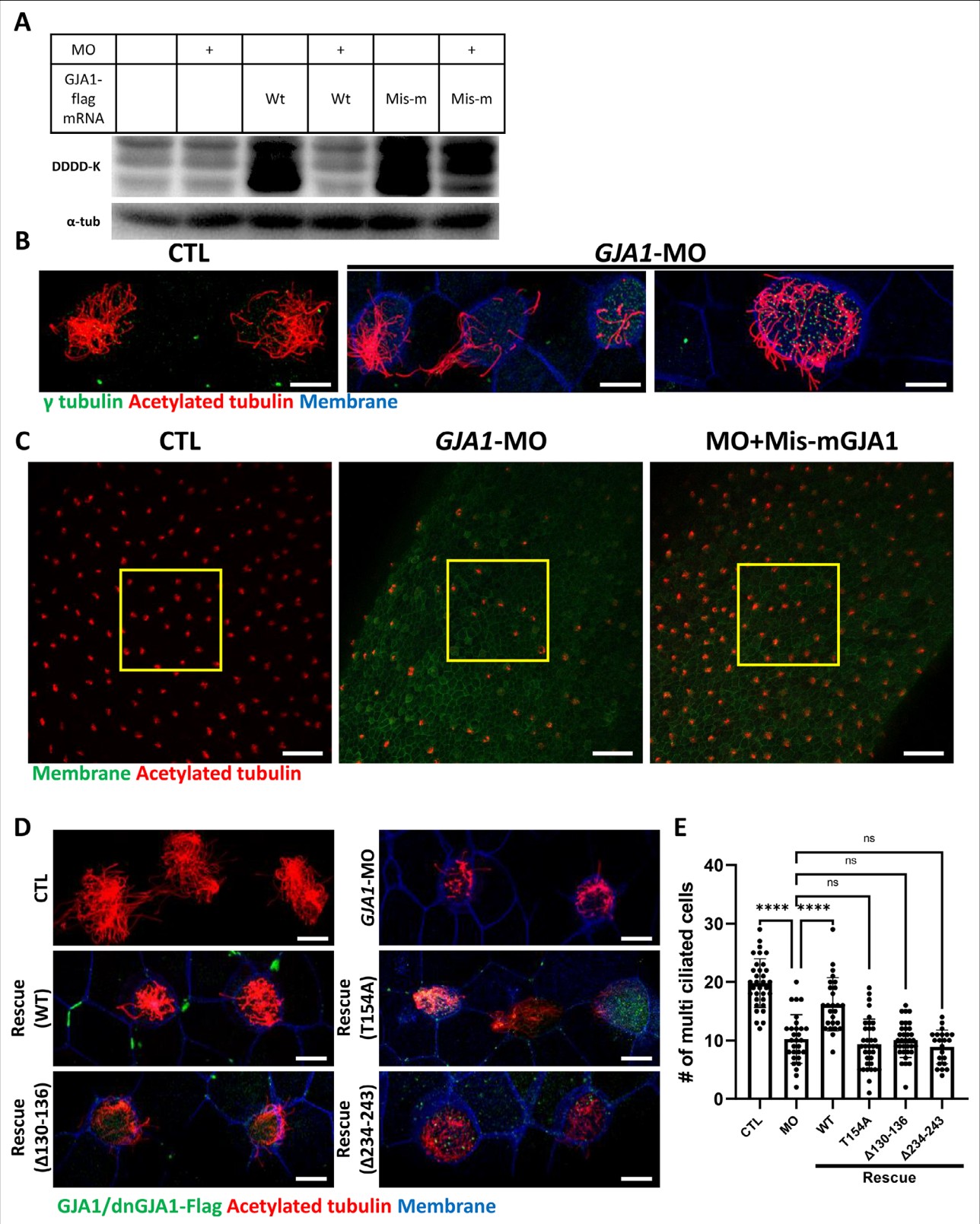

**Figure 3.** Morpholino-mediated knockdown of *GJA1* causes severe disruption of ciliogenesis in *Xenopus* multiciliated epithelial cells. (**A**) Wild-type (WT) *GJA1* (Wt-*GJA1*-Flag) mRNA or silent mutant mRNA (Mis-m*GJA1*-Flag; non-targetable by *GJA1*-MO), which was modified to be mismatched to the morpholino oligo (MO) sequence, was co-injected with *GJA1*-MO into embryos. Immunoblotting was performed with an anti-Flag antibody for GJA1-Flag or with α-tubulin as the loading control. *GJA1*-MO efficiently inhibited GJA1-Flag expression from Wt-*GJA1* mRNA. (**B**) Microinjection of

*Figure 3 continued on next page*

*Figure 3 continued*

a *GJA1* antisense morpholino (*GJA1*-MO) disrupted normal ciliary axoneme assembly in multiciliated cells. The ciliary axonemes were labeled with an acetylated tubulin antibody (red), and centrosomes or basal bodies were stained with a γ-tubulin antibody (green). The cell membranes were labeled by microinjection of membrane RFP and detected using an anti-RFP antibody (blue). Scale bars: 10 μm. (**C**) Low-magnification confocal images of *Xenopus* multiciliated epithelial cells. Embryos were either injected with *GJA1*-MO alone or co-injected with MO-mismatched-*GJA1* mRNA (Mis-m*GJA1*) and *GJA1*-MO. Mis-m*GJA1* mRNA effectively rescued the ciliary defects. Multi-cilia bundles were stained with acetylated tubulin (red), and membranes were labeled by microinjection of membrane-GFP and detected with an anti-GFP antibody (green). Scale bars: 100 μm. (**D**) Flag-tagged wild-type or dominant-negative *GJA1* mutant mRNAs were co-injected with *GJA1*-MO. All mutant mRNAs contained MO-mismatched target sequences. Only WT-*GJA1*-Flag mRNA rescued the loss-of-function phenotype. Each GJA1 flag-tagged protein was detected with a flag antibody (green). Ciliary axonemes (red) and cell membranes (blue) were visualized with an anti-acetylated antibody and an anti-GFP antibody, respectively. Scale bars: 10 μm. (**E**) Statistical analysis of the number of multiciliated cells per unit area in panels (**C**) and (**D**). Error bars represent the mean ± SD. P values were determined with the ordinary one-way ANOVA (p****<0.0001, Pⁿˢ >0.05). n=33(CTL), 29(MO), 26(Rescue(WT)), 35(Rescue(T154A)), 34(Rescue(Δ130-136)), 22(Rescue(Δ234-243)). Raw values are provided in the *Figure 3—source data 1* file.

The online version of this article includes the following source data and figure supplement(s) for figure 3:

**Source data 1.** The number of multiciliated cells per unit area.

**Source data 2.** The uncropped images of Western blots.

**Figure supplement 1.** Morpholino oligo (MO)-mediated knockdown of *GJA1* does not affect the cell fate specification of multiciliated cells.

**Figure supplement 1—source data 1.** The number of *DNAH9*-positive cells per unit area.

**Figure supplement 2.** CRISPR/Cas9-mediated *GJA1* knockout shows ciliary defect phenotypes similar to morpholino oligo (MO)-mediated knockdown.

**Figure supplement 2—source data 1.** The number of multiciliated cells per unit area.

**Figure supplement 2—source data 2.** The uncropped images of DNA gel electrophoresis.

---

*Figure 4C and D*, *Figure 4—videos 1 and 2*). Furthermore, expression of the left LPM-specific marker *PITX2* was abnormal in GJA1-morphant embryos (*Figure 4E and F*). These data demonstrate that GJA1 is also involved in motile nodal cilium formation in the GRP in *Xenopus*.

## siRNA-mediated knockdown of *GJA1* disrupts primary ciliogenesis in human RPE1 cells

Because knockdown of *GJA1* expression caused severe defects in the motile cilia of *Xenopus* epithelial tissues, we next examined the functions of GJA1 in primary ciliogenesis in human RPE1 cells. We first confirmed that low-serum starvation effectively initiated cilia formation in human RPE1 cells (*Figure 5—figure supplement 1A,B*). Then, we depleted *GJA1* in human RPE1 cells by siRNA transfection, which was confirmed by immunoblotting (*Figure 5—figure supplement 1C*). Next, we observed primary cilium formation after inducing ciliogenesis under low-serum starvation conditions. Indeed, GJA1 depletion significantly disrupted cilia formation in RPE1 cells compared to control cells. Additionally, co-transfection of an siRNA-non-targetable-*GJA1* cDNA plasmid with *GJA1* siRNA partially rescued the siRNA phenotypes (*Figure 5A and B*). Next, we observed the localization of GJA1 in human RPE1 cells by performing an immunofluorescence analysis. As previously shown, GJA1 protein was localized to the gap junctions at the cell periphery (*Figure 5A*, yellow square). Unexpectedly, however, we did not observe ciliary localization of GJA1 in the primary cilium. GJA1 signals accumulated in pericentriolar regions, where the primary cilium originates (*Figure 5C*, white arrow). We further analyzed the subcellular localization of GJA1 by co-immunostaining GJA1 with the trans-Golgi marker TGN46 or the pericentriolar material (PCM) marker BBS4 and observed that GJA1 was broadly distributed in the Golgi and pericentriolar regions (*Figure 5—figure supplement 2A*). Interestingly, acetylated microtubules in the pericentriolar region were noticeably reduced in GJA1-depleted cells (*Figure 5A*) but the Golgi structure were not affected by GJA1 depletion (*Figure 5—figure supplement 2B*). From this observation, we speculated that GJA1 plays a role in primary cilium formation by affecting trafficking around the pericentriolar regions or PCM, which is involved in cilia formation (*Moser et al., 2010*).

Next, we examined the subcellular localization of several key proteins that are critical for ciliogenesis under GJA1-depleted conditions. Immunofluorescence staining of endogenous Arl13b, BBS4, CP110, IFT20, IFT88, and TTBK2, which are well-known PCM or ciliary markers (*Figure 6—figure supplement 1A*), was performed. BBS4 and CP110 showed abnormal distributions in *GJA1* siRNA-transfected cells (*Figure 6A–C*), and no significant differences were found in the expression of other

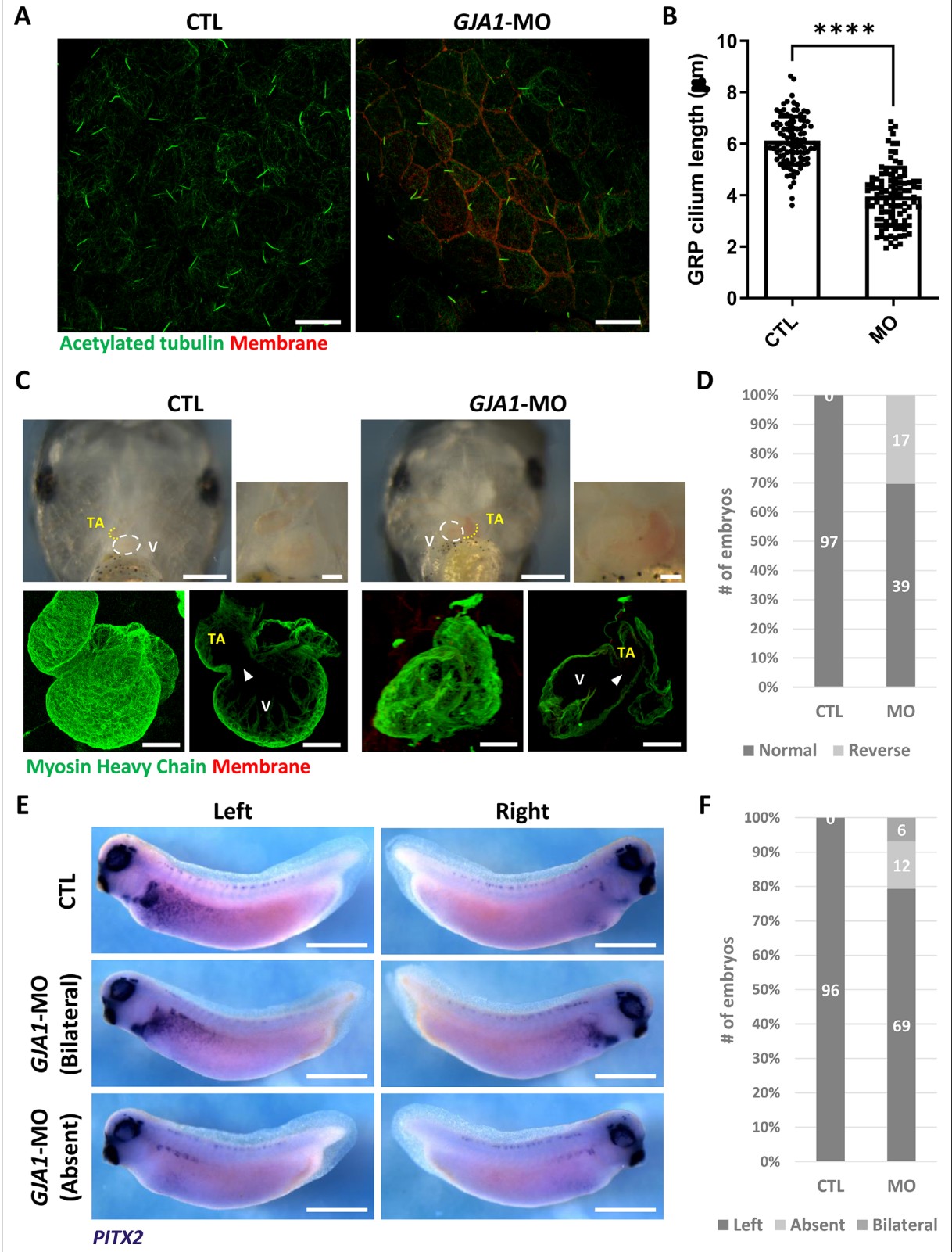

**Figure 4.** GJA1 depletion disrupts GRP cilium formation and left-right asymmetry. (**A**) *GJA1*-MO was injected into the dorso-vegetal blastomeres to target gastrocoel roof plate (GRP) tissues. Nodal cilium in the GRP were stained with an anti-acetylated tubulin antibody (green), and the cell membrane was visualized by membrane-GFP, which was detected by an anti-GFP antibody (red). Scale bars: 20 μm. (**B**) The average lengths of GRP ciliary axonemes were plotted. Error bars represent the mean ± SD. P values were determined with the two-tailed t-test (p****<0.0001). n=100(CTL, MO). Raw

*Figure 4 continued on next page*

*Figure 4 continued*

values are provided in the ***Figure 4—source data 1*** file. (**C–D**) GJA1 depletion reversed heart looping in *Xenopus* embryos. Embryonic hearts were visualized by immunostaining with an anti-myosin heavy chain antibody (lower panel). GJA1 morphants displayed reversed heart looping. Approximately 30% of MO-injected embryos displayed reversed heart looping. Raw values are provided in the ***Figure 4—source data 2*** file. Scale bars: 500 µm (upper-left panel), 100 µm (upper-right, lower panel). TA; truncus arteriosus, V; ventricle. (**E–F**) Expression of the left LPM-specific marker *PITX2* was disrupted in GJA1 morphants. Raw values are provided in the ***Figure 4—source data 3*** file. Scale bars: 1 mm.

The online version of this article includes the following video and source data for figure 4:

**Source data 1.** The average lengths of GRP ciliary axonemes.

**Source data 2.** The numbers of embryos displaying reversed heart looping.

**Source data 3.** The numbers of embryos displaying normal or abnormal *PITX2* expression.

**Figure 4—video 1.** Heartbeat of a wildtype embryo.

https://elifesciences.org/articles/81016/figures#fig4video1

**Figure 4—video 2.** Heartbeat of GJA1 a morphant embryo.

https://elifesciences.org/articles/81016/figures#fig4video2

proteins, such as Arl13b, IFT20/88 and TTBK2 (*Figure 6—figure supplement 1B*). During cell cycle progression, CP110 is localized at the cap of mother and daughter centrioles and inhibits ciliary axoneme elongation. For proper primary cilium formation, CP110 must be removed from the cap of the mother centriole to initiate elongation of the ciliary axonemes (*Tsang et al., 2008*; *Tsang and Dynlacht, 2013*). In addition, BBS4, which is a component of the BBSome complex, is localized at the PCM and interacts with PCM-1, dynein, and intraflagellar transport protein (IFT) during the delivery of ciliary building blocks to ciliary axonemes (*Figure 6—figure supplement 1A*; *Uytingco et al., 2019*; *Nachury et al., 2007*). Interestingly, we frequently observed two CP110 spots in both mother and daughter centrioles in *GJA1* siRNA-transfected RPE1 cells (*Figure 6A and B*). In addition, BBS4 was delocalized from the PCM and basal bodies (*Figure 6C*) with non-ciliogenesis phenotypes. Although the mechanism of CP110 during *Xenopus* multiciliogenesis has not been determined, CP110 phenotypes were similar to the GJA1 loss-of-function phenotype in RPE1 cells. GJA1-depleted embryos showed higher CP110 signal levels than those of wild-type embryos (*Figure 6—figure supplement 2A*, B). These data suggest that GJA1 is critical for primary cilium formation and may function in the cytoplasmic assembly of ciliogenesis factors around the pericentriolar region.

## Rab11 interacts with GJA1, and knockdown of *GJA1* interrupts Rab11 trafficking to basal bodies during ciliogenesis

To identify the putative binding and functional partners of GJA1, we performed GJA1 IP-MS. Then, we performed gene ontology term analysis with 232 detected proteins and identified two major clusters of interest, regulation of $Ca^{2+}$ and ciliogenesis (*Figure 7—figure supplement 1*, *Supplementary file 1*). A recent study revealed that gap junctions, especially those containing GJA1, mediate intracellular $Ca^{2+}$ waves and regulate ependymal multicilia maintenance through the Wnt-GJA1-$Ca^{2+}$ axis (*Zhang et al., 2020*). Therefore, we determined whether GJA1 depletion compromises cilia formation by affecting the intracellular $Ca^{2+}$ concentration in RPE1 cells. GJA1 depletion or overexpression resulted in similar $Ca^{2+}$ concentrations in the resting state and slightly affected $Ca^{2+}$ entry compared with that in control RPE1 cells (*Figure 7—figure supplement 2A*, B). However, the difference in $Ca^{2+}$ entry was not as significant as that in previously reported studies. Next, we focused on the clusters of ciliogenesis. Among dozens of interacting ciliary proteins, we selected Rab8a and Rab11a as candidates (*Figure 7—figure supplement 1*). Both Rab8 and Rab11 have previously been reported to be involved in ciliogenesis (*Knödler et al., 2010*; *Westlake et al., 2011*; *Nachury et al., 2007*; *Blacque et al., 2018*). Rab8 is localized in the primary cilium and regulates ciliary membrane extension (*Nachury et al., 2007*; *Blacque et al., 2018*). Previous studies have also shown that Rab8 interacts with CP110 and BBS4 (*Tsang et al., 2008*; *Tsang and Dynlacht, 2013*; *Nachury et al., 2007*). Rab11 is enriched at the basal part of the primary cilium and recruits Rab8 to the basal body by interacting with Rabin8 (*Westlake et al., 2011*; *Blacque et al., 2018*). Additionally, during basal body maturation, Rab11-positive membrane vesicles are transported to the distal appendages of basal bodies and form large ciliary vesicles. This distal appendage vesicle assembly that forms ciliary vesicles is a prerequisite for CP110 removal and promotes the uncapping of the mother centrioles, which is followed by the

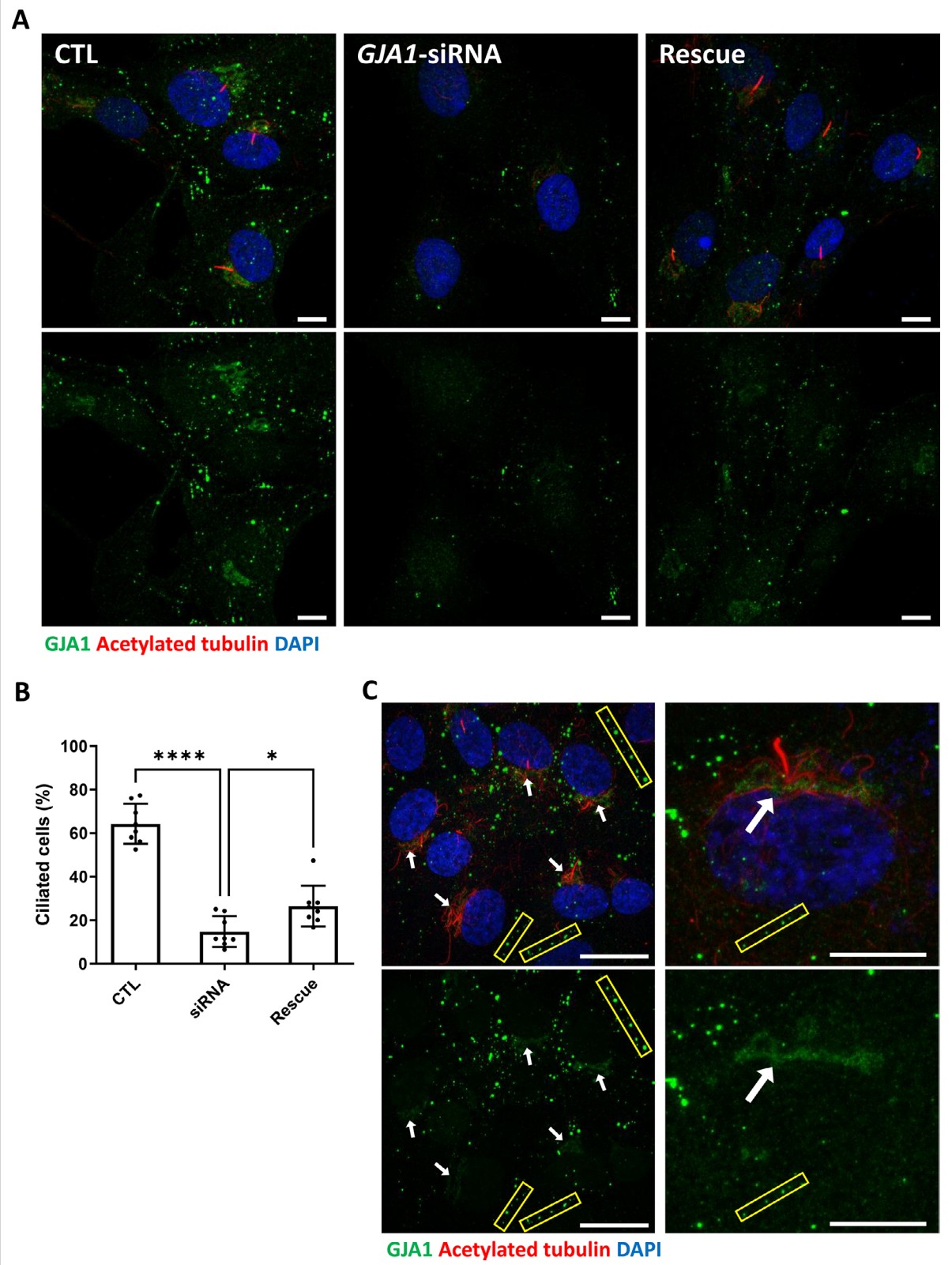

GJA1 **Acetylated tubulin** DAPI

**Figure 5.** siRNA-mediated knockdown of *GJA1* disrupts primary cilium formation in human RPE1 cells. (**A**) RPE1 cells were serum-starved for 24 hr and labeled with an anti-GJA1 antibody (green), an anti-acetylated tubulin antibody (red), and DAPI (blue). siRNA-mediated knockdown of *GJA1* inhibited primary cilium formation, and the ciliary defects in human RPE1 cells were rescued by *GJA1* transfection. Scale bars: 10 μm. (**B**) Statistical analysis of the percentage of ciliated cells in panel (**A**). Error bars represent the mean ± SD. P values were determined with the ordinary one-way ANOVA

*Figure 5 continued on next page*

*Figure 5 continued*

(p****<0.0001, p*=0.0238). n=8(CTL, siRNA, Rescue), cell n=417(CTL), 242(siRNA), 193(Rescue). Raw values are provided in the *Figure 5—source data 1* file. (**C**) RPE1 cells were serum-starved for 24 hr and labeled with an anti-GJA1 antibody (green), an anti-acetylated tubulin antibody (red), and DAPI (blue). GJA1 was localized around the pericentriolar region in addition to gap junctions. Scale bars: 20 μm (left panel), 10 μm (right panel).

The online version of this article includes the following source data and figure supplement(s) for figure 5:

**Source data 1.** The percentage of ciliated cells.

**Figure supplement 1.** Ciliation of RPE1 cells by serum starvation and the efficacy of *GJA1* siRNA.

**Figure supplement 1—source data 1.** The percentage of ciliated cells.

**Figure supplement 1—source data 2.** The uncropped images of Western blots.

**Figure supplement 2.** GJA1 localizes to the Golgi complex but GJA1 depletion does not affect Golgi morphology.

recruitment of intraflagellar transport proteins and transition zone proteins for ciliogenesis (*Sánchez and Dynlacht, 2016*; *Lu et al., 2015*).

We first confirmed the interaction between GJA1 and Rab8a or Rab11a by performing co-immuno-precipitation, followed by a western blot assay. Consistent with the IP-MS data, GJA1 was successfully immunoprecipitated with Rab8a (*Figure 7—figure supplement 3A*) or Rab11a (*Figure 7A*), whereas the rabbit IgG negative control did not show any signals. Next, we confirmed the localization of GJA1 and Rab11 during ciliogenesis. As expected, GJA1 and Rab11 were partially co-localized around the pericentriolar region and acetylated tubulin network (*Figure 7B*). In particular, GJA1 surrounded the Rab11 cluster, which was accumulated around the basal bodies. Furthermore, we observed decreased Rab8a and Rab11 protein levels in GJA1-depleted RPE1 cells (*Figure 7—figure supplement 3B*). Additionally, the accumulation of Rab11 at the base of the primary cilium was absent upon GJA1 depletion (*Figure 7C–F*). These phenotypes were partially rescued with *GJA1*-cloned plasmid co-transfected cells (*Figure 7C and D*). We further confirmed the presence of Rab11-positive ciliary vesicles using structured illumination microscopy (SIM). At higher resolutions, SIM analysis showed that Rab11-positive vesicles encircled the base of the ciliary axoneme. In contrast, GJA1-depleted cells showed scattered Rab11 signals (*Figure 7E and F*). We next investigated how dnGJA1 mutants may disturb ciliogenesis by performing co-immunoprecipitation with dnGJA1 proteins and Rab11a. Interestingly, all dnGJA1 mutant proteins failed to interact with Rab11a (*Figure 8A and B*).

Overall, these data suggest that GJA1 controls the trafficking of Rab11-positive vesicles to the base of cilia for ciliogenesis.

## Discussion

The gap junction complex forms a channel that mediates cell-to-cell communication. Gap junction-mediated cell-to-cell communication is indispensable for normal embryonic development, and disruption of gap junction complex components causes embryonic lethality and various phenotypes in multiple organs (*Meşe et al., 2007*; *Laird, 2010*; *Dobrowolski and Willecke, 2009*; *Pfenniger et al., 2011*; *Srinivas et al., 2018*). For example, mutation of *GJA1*, which is a major gap junction complex component, causes a congenital disorder, ODDD, in humans (*Paznekas et al., 2003*). Interestingly, recent studies have strongly suggested that GJA1 exerts diverse functions, in addition to gap junction formation, to control cellular migration and polarity via regulation of cytoskeleton components, such as actin filaments and microtubules, in various cell types, including cancer cells, neuronal progenitors, and cardiac neural crest cells (*Matsuuchi and Naus, 2013*; *Rhee et al., 2009*; *Francis et al., 2011*; *Crespin et al., 2010*). Additionally, the non-channel functions of GJA1 have been suggested to explain the complex disease phenotypes caused by GJA1 mutations in mice and humans (*Pfenniger et al., 2011*; *Laird, 2014*; *Laird, 2008*).

While we were examining the molecular mechanism of GJA1 in ciliogenesis, *Zhang et al., 2020* published a paper arguing that GJA1 participates in the maintenance of ependymal motile cilia in the zebrafish spinal cord and in mouse models. In this publication, Zhang et al., argued that GJA1 mediates the $Ca^{2+}$ wave and signaling in the Wnt-PLC-IP3 cascade to maintain motile cilia.

In this study, we confirmed GJA1 localization around the pericentriolar region and ciliary axonemes for the first time, suggesting a direct function of GJA1 in ciliogenesis. Moreover, we confirmed that GJA1 depletion specifically affected CP110 removal from the mother centrioles and caused

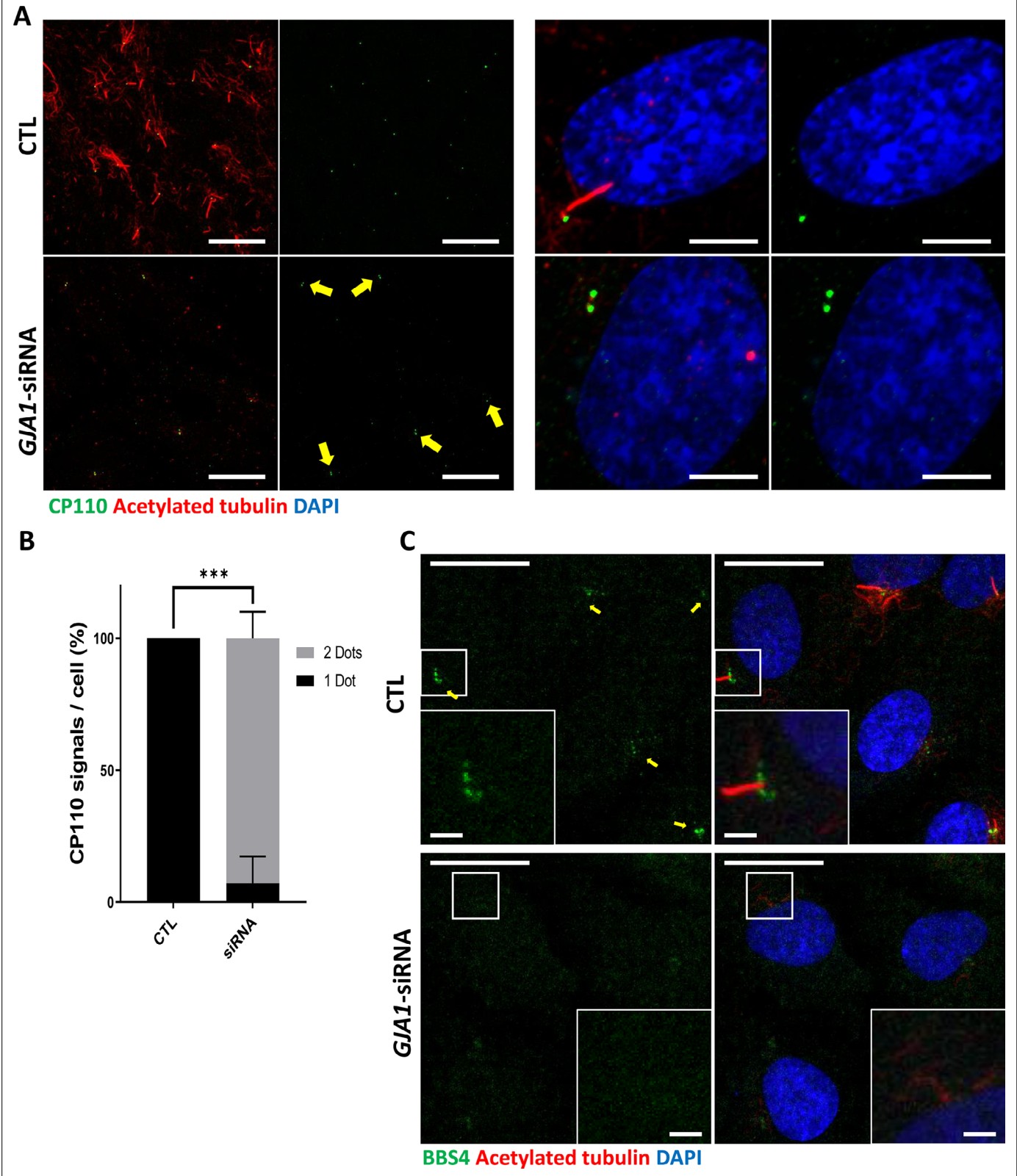

**Figure 6.** Knockdown of *GJA1* results in the abnormal distribution of pericentriolar proteins involved in ciliogenesis. (**A**) RPE1 cells were transfected with *GJA1* siRNA and immunostained with an anti-CP110 antibody (green), an anti-acetylated tubulin antibody (red), and DAPI (blue). Two CP110 spots in both mother and daughter centrioles were observed in GJA1-depleted RPE1 cells. Scale bars: 20 μm (left panel), 5 μm (right panel). (**B**) The percentage of CP110 dots per cell in panel (**A**). Error bars represent the mean ± SD. P values were determined with the with the two-way ANOVA (p***=0.0004).

*Figure 6 continued on next page*

*Figure 6 continued*

n=2(CTL, siRNA/1 Dot, 2Dots), cell n = 24(CTL), 22 (siRNA). Raw values are provided in the *Figure 6—source data 1* file. (**C**) RPE1 cells were transfected with *GJA1* siRNA and immunostained with an anti-BBS4 antibody (green) and an anti-acetylated tubulin antibody (red), and DAPI (blue). BBS4 puncta in the pericentriolar material (PCM) were absent from the pericentriolar region of GJA1-depleted cells. The pericentriolar region is shown with higher magnification in the inset panels. Scale bars: 20 µm, 5 µm (inset).

The online version of this article includes the following source data and figure supplement(s) for figure 6:

**Source data 1.** The percentage of CP110 dots per cell.

**Figure supplement 1.** Subcellular localization of ciliary proteins near the pericentriolar region in GJA1-depleted RPE1 cells.

**Figure supplement 2.** CP110 localization in multiciliated cells in *GJA1*-MO-injected *Xenopus* embryos.

**Figure supplement 2—source data 1.** The relative intensity of CP110 signals in the ciliated cells.

delocalization and BBS4-positive vesicle around the pericentriolar region; however, several other key ciliary and PCM molecules, such as Arl13b, IFT20, IFT88, and TTBK2, were not significantly affected. These data suggest that GJA1 may perform specific functions in general ciliogenesis, in addition to functioning as a mediator of the $Ca^{2+}$ wave, for the maintenance of motile cilia.

Furthermore, in IP-MS assays, we found that Rab11 interacted with GJA1, which can explain how GJA1 controls CP110 removal from the mother centriole for ciliogenesis. Rab11 is a major regulator of ciliary vesicle trafficking and is necessary for ciliary vesicle accumulation on the mother centriole, which is suggested to be a prerequisite for removal of CP110 from the mother centriole (*Sánchez and Dynlacht, 2016*; *Lu et al., 2015*). Finally, we showed that the dnGJA1 mutant proteins in this study failed to interact with Rab11a, which may explain how GJA1 controls cilia formation. Our findings strongly support that GJA1 controls trafficking and accumulation to Rab11/8 positive vesicles in the mother centriole and hence facilitates removal of CP110 to initiate cilia formation and this process is probably necessary pericentriolar microtubule networks (*Figure 8C*).

Previous studies have investigated the possible function of gap junction proteins, such as GJB2 (gap junction protein β2; CX26), GJA1 (CX43), and GJA7 (gap junction protein α7; CX43.4), in determining left-right asymmetry. Most studies on the function of gap junction proteins in determining laterality have indicated that the channel function is critical for mediating leftward signals, which are driven by nodal cilia (*Beyer et al., 2012*; *Levin and Mercola, 1998*; *Hatler et al., 2009*). For example, GJB2 is a key gap junction protein involved in determining left-right asymmetry, whereas the GRP cilium is largely unaffected by GJB2 depletion (*Beyer et al., 2012*).

Our study supports that GJA1, a major connexin (CX43), is critical for formation of both motile and primary cilia. The cytosolic domain of GJA1 contains a microtubule-binding motif that is necessary for trafficking to the gap junction complex. Furthermore, GJA1 seems to exhibit gap junction-independent functions that mainly control the cytoskeleton and provide directional cues for migrating cells (*Giepmans et al., 2001*; *Francis et al., 2011*; *Fu et al., 2017*). Interestingly, cilia have also been shown to be necessary for the polarization of migrating cells and directional protrusions, such as lamellipodia (*Veland et al., 2014*; *Schneider et al., 2010*). Given the complex loss-of-function phenotype of GJA1, it is feasible to hypothesize that some gap-junction-independent functions of GJA1 may be mediated by regulating cilia formation. GJA1 was localized around the pericentriolar region in human RPE1 cells, and GJA1 depletion resulted in reduced acetylated tubulin levels in pericentriolar microtubules. Pericentriolar microtubules are critical for the transport of ciliary building blocks and are involved in cilia formation by organizing ciliary protein transport (*Kim et al., 2008*; *Kim et al., 2012*; *Lopes et al., 2011*). Further studies to examine the molecular functions of GJA1 in transporting ciliary building blocks may elucidate the molecular function of GJA1 in cilia formation.

# Materials and methods

**Key resources table**

| Reagent type (species) or resource | Designation | Source or reference | Identifiers | Additional information |
|---|---|---|---|---|
| Strain, strain background (*Xenopus laevis*) | *Xenopus; Xenopus laevis* | Korea National Research Resource Center (KNRRC) | KXRCR000001; KXRCR000002 | Materials and Methods 1. |
| Gene (*Xenopus laevis*) | GJA1 | Xenbase | XB-GENE-876609 | Materials and Methods 4. |
| Gene (*Homo sapiens*) | GJA1 | NCBI | Gene ID: 2697 | Materials and Methods 4. |
| Gene (*Homo sapiens*) | Rab11a | NCBI | Gene ID: 8766 | Materials and Methods 4. |
| Cell line (*Homo sapiens*) | hTERT-RPE1 | ATCC | Cat# CRL-4000; RRID:CVCL_4388 | Materials and Methods 2. |
| Sequence-based reagent | GJA1-MO | Gene Tools | XB-GENE-876609 | Materials and Methods 1.2. |
| Sequence-based reagent | GJA1-siRNA | Genolution | Gene ID: 2697 | Materials and Methods 2.1. |
| Antibody | Anti-CP110 (rabbit polyclonal) | Proteintech | 12780–1-AP; RRID:AB_10638480 | Materials and Methods 5. |
| Antibody | Anti-BBS4 (rabbit polyclonal) | Proteintech | 12766–1-AP; RRID:AB_10596774 | Materials and Methods 5. |
| Antibody | Anti-Rab11 (rabbit monoclonal) | Cell Signaling | #5589; RRID:AB_10693925 | Materials and Methods 5, 6. |
| Antibody | Anti-GJA1 (rabbit polyclonal) | ThermoFisher | PA1-25098; RRID:AB_779905 | Materials and Methods 5, 6, 7. |
| Antibody | Anti-GJA1 (mouse monoclonal) | ThermoFisher | 13–8300; RRID:AB_2533038 | Materials and Methods 5. |

### *Xenopus laevis* rearing

Adult female and male *Xenopus laevis* were provided by the Korea National Research Resource Center (KNRRC, KXRCR000001, KXRCR000002). Ovulation was induced in adult female *Xenopus laevis* by injection of human chorionic gonadotropin. Eggs were then fertilized in vitro and dejellied with 3% cysteine in 1/3×Marc's Modified Ringers (MMR) (pH 7.8–7.9) solution. After fertilization, eggs were reared in 1/3×MMR solution.

### Microinjection procedure

For microinjections, embryos were placed in 1% Ficoll in 1/3×MMR solution. To target the epithelial or GRP tissue in *Xenopus* embryos, we injected into the ventral-animal region (epithelial tissue) or dorsal-vegetal region (GRP tissue) of blastomeres in the two- or four-cell stage (*Faber and Nieuwkoop, 1967*) using a Picospritzer III microinjector (Parker, NH, USA). The injected eggs were grown in 1/3×MMR with an antibiotic (gentamycin). We injected 80 ng of MO per embryo, and wild-type or dominant-negative mutant *GJA1* mRNAs were injected in various amounts.

### Morpholino

We designed a translation-blocking antisense MO for *GJA1* based on the sequence from the Xenbase database. The MO was manufactured by Gene Tools (OR, USA) (*GJA1*-MO: 5′-TTCCTAAGGCACT CCAGTCACCCAT-3′).

### Fixation

To analyze epithelial cilia development, we fixed embryos at stage 26–30 using MEMFA (2×MEM salts and 7.5% formaldehyde) or Dent's fixative solution (20% DMSO in methanol).

### Cilia isolation

Cilia isolation was performed using a modified $Ca^{2+}$ shock method, as previously described (*Hastie et al., 1986*). Briefly, deciliation solution (20 mM Tris-HCl [pH 7.5], 50 mM NaCl, 10 mM $CaCl_2$, 1 mM EDTA, 7 mM β-mercaptoethanol, and 0.1% Triton X-100) was added to the embryos, and the test tube

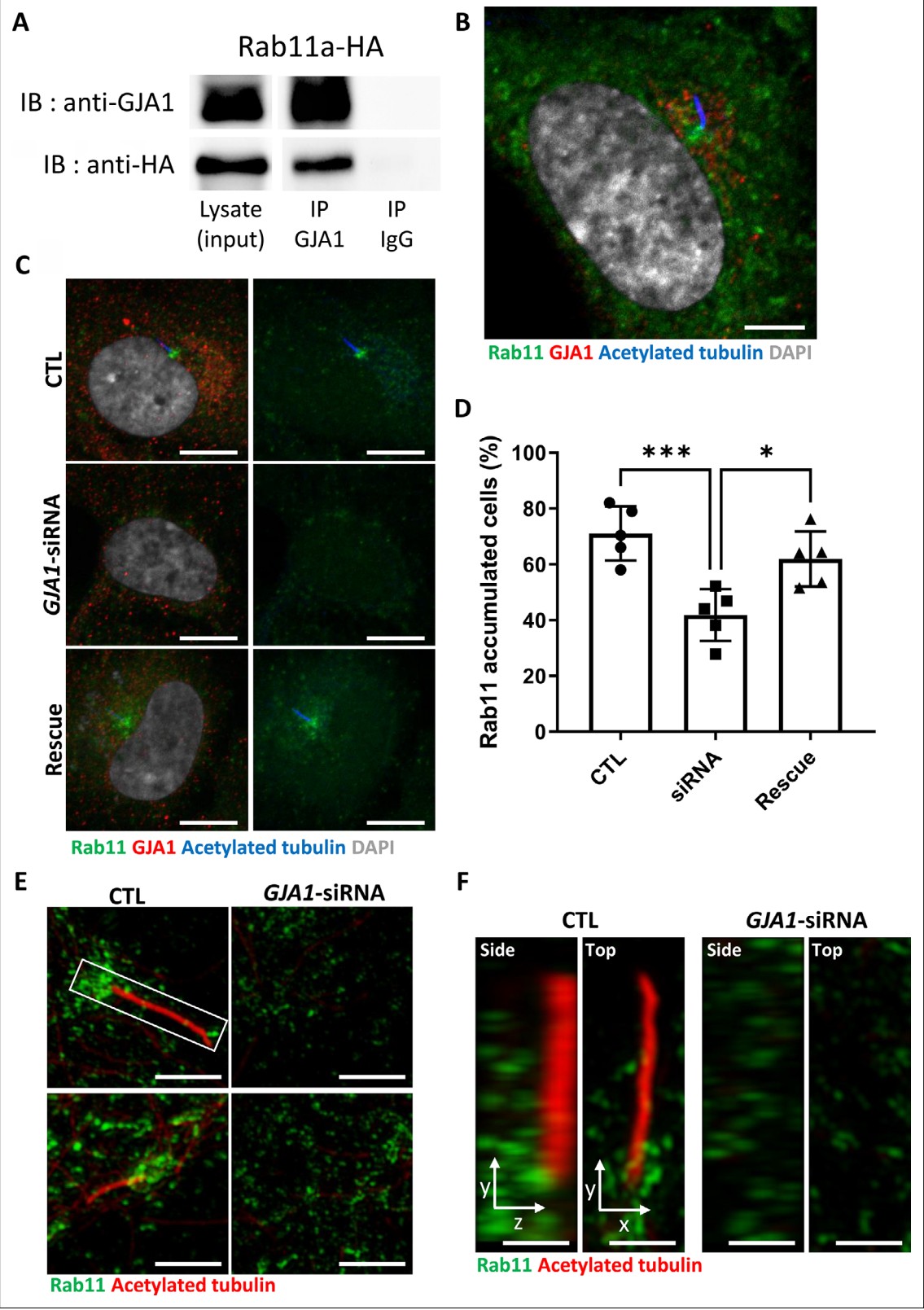

**Figure 7.** Knockdown of *GJA1* disrupted Rab11 trafficking to the ciliary base. (**A**) HA-tagged *Rab11a* was expressed in RPE1 cells, and the lysates were used for immunoprecipitation assays with GJA1 antibody-conjugated or IgG-conjugated beads. The immunoprecipitants were analyzed by immunoblotting with anti-GJA1 or anti-HA antibodies. Rab11a-HA was co-immunoprecipitated with endogenous GJA1. (**B**) RPE1 cells were serum-starved for 24 hr, and immunofluorescence staining analysis was performed with an anti-Rab11 antibody (green), an anti-GJA1 antibody (red), an

*Figure 7 continued on next page*

*Figure 7 continued*

anti-acetylated tubulin antibody (blue), and DAPI (gray). Rab11 and GJA1 accumulated around the pericentriolar-ciliary base. Scale bars: 5 µm. (**C**) *GJA1*-siRNA-treated RPE1 cells were immunostained with an anti-Rab11 antibody (green), an anti-GJA1 antibody (red), an anti-acetylated tubulin antibody (blue), and DAPI (gray). siRNA-mediated knockdown of *GJA1* inhibited Rab11 accumulation around the pericentriolar region and basal bodies, and *GJA1* transfection rescued the phenotype in RPE1 cells. Scale bars: 10 µm. (**D**) Statistical analysis of Rab11 accumulation in panel (**C**). Error bars represent the mean ± SD. P values were determined by one-way ANOVA (p***=0.0008, p*=0.0117). n=6(CTL, siRNA, Rescue), cell n=221(CTL), 157(siRNA), 133(Rescue). Raw values are provided in the *Figure 7—source data 1* file. (**E–F**) Super-resolution images of Rab11 localization. RPE1 cells were labeled with an anti-Rab11 antibody (green) and an anti-acetylated tubulin antibody (red). GJA1 depletion was not observed in Rab11-positive ciliary vesicles in the pericentriolar region. Images were obtained by SIM. Scale bars: 5 µm (**E**), 1 µm (**F**).

The online version of this article includes the following source data and figure supplement(s) for figure 7:

**Source data 1.** The percentage of Rab11 accumulated cell.

**Source data 2.** The uncropped images of western blots.

**Figure supplement 1.** Identification of GJA1-interacting proteins by IP-MS.

**Figure supplement 2.** Basal $Ca^{2+}$ levels and $Ca^{2+}$ influx in GJA1-depleted RPE1 cells.

**Figure supplement 2—source data 1.** Basal $Ca^{2+}$ levels in RPE1 cells.

**Figure supplement 2—source data 2.** $Ca^{2+}$ influx in RPE1 cells.

**Figure supplement 3.** GJA1 interacts with Rab8a.

**Figure supplement 3—source data 1.** The uncropped images of western blots.

**Figure supplement 3—source data 2.** The uncropped images of western blots.

---

was gently inverted for 30 s. To remove the cellular debris, samples were centrifuged at 1500 g for 2 min, and the supernatants were transferred to new e-tubes. Cilia in the supernatant were pelleted by centrifugation at 12,000 g for 5 min. After discarding the supernatant, MEMFA or Dent's fixative solution was added immediately to fix the cilia pellet.

## In situ hybridization

Whole-mount in situ hybridization was performed with a modified method, as previously described (*Harland, 1991*). The antisense probe was transcribed from the approximately 800 bp amplified PCR product of *PITX2* (Primer; Sense: 5'-CGCAAACTGGTGACAACCTGTG-3', SP6-antisense: 5'-gcgatttag gtgacactatagTGCCAGGCTGGAGTTACATGTG-3') or *DNAH9* (Primer; Sense: 5'-ACAGGCTGGTGCT GCAGGA-3', SP6-antisense: 5'-gcgatttaggtgacactatagCAAAATGACGCTGGAGGGG-3') using the mMESSAGE mMACHINE SP6 Transcription Kit (Invitrogen, ThermoFisher Scientific, MA, USA) and a dig-dNTP mix (Roche, Basel, Switzerland).

## GRP tissue explant preparation

GRP tissue explants were prepared as previously described (*Blum et al., 2009*). To image GRP nodal cilium, stage 17 embryos were fixed and sliced at the anterior head before immunofluorescence staining. Then, GRP tissue explants were mounted on a cover glass.

## Genome editing with CRISPR/Cas9 technology

The *Xenopus laevis GJA1* crRNA sequence was designed using the CHOPCHOP website. The designed *GJA1* crRNA (Sense: 5'-GTCTGCAATACTCAGCAACCagg-3') and Alt-R CRISPR-Cas9 crRNA and the tracrRNAs were purchased from IDT. *GJA1* CRISPR/Cas9 was prepared using the protocol provided by the manufacturer. spCas9 protein with guide RNA (RNP) was injected into the ventral-animal region of the blastomeres at the two-cell stage. We injected 20 fmol RNP in each embryo. Genomic DNA was extracted from stage 28 embryos to confirm *GJA1* CRISPR/Cas9 efficiency. Genomic DNA was used in PCR amplification with specific primer pairs, which targeted the vicinity of the guide RNA sequence. The amplified PCR product was used for in vitro CRISPR/Cas9 experiments or sequencing analysis (Bionics, Seoul, Republic of Korea).

## Human RPE1 cell culture

hTERT-RPE-1 cells were obtained from ATCC (VA, USA, CRL-4000) and were cultured under standard mammalian cell conditions with 10% FBS (Gibco, ThermoFisher Scientific) and 1% antibiotics in

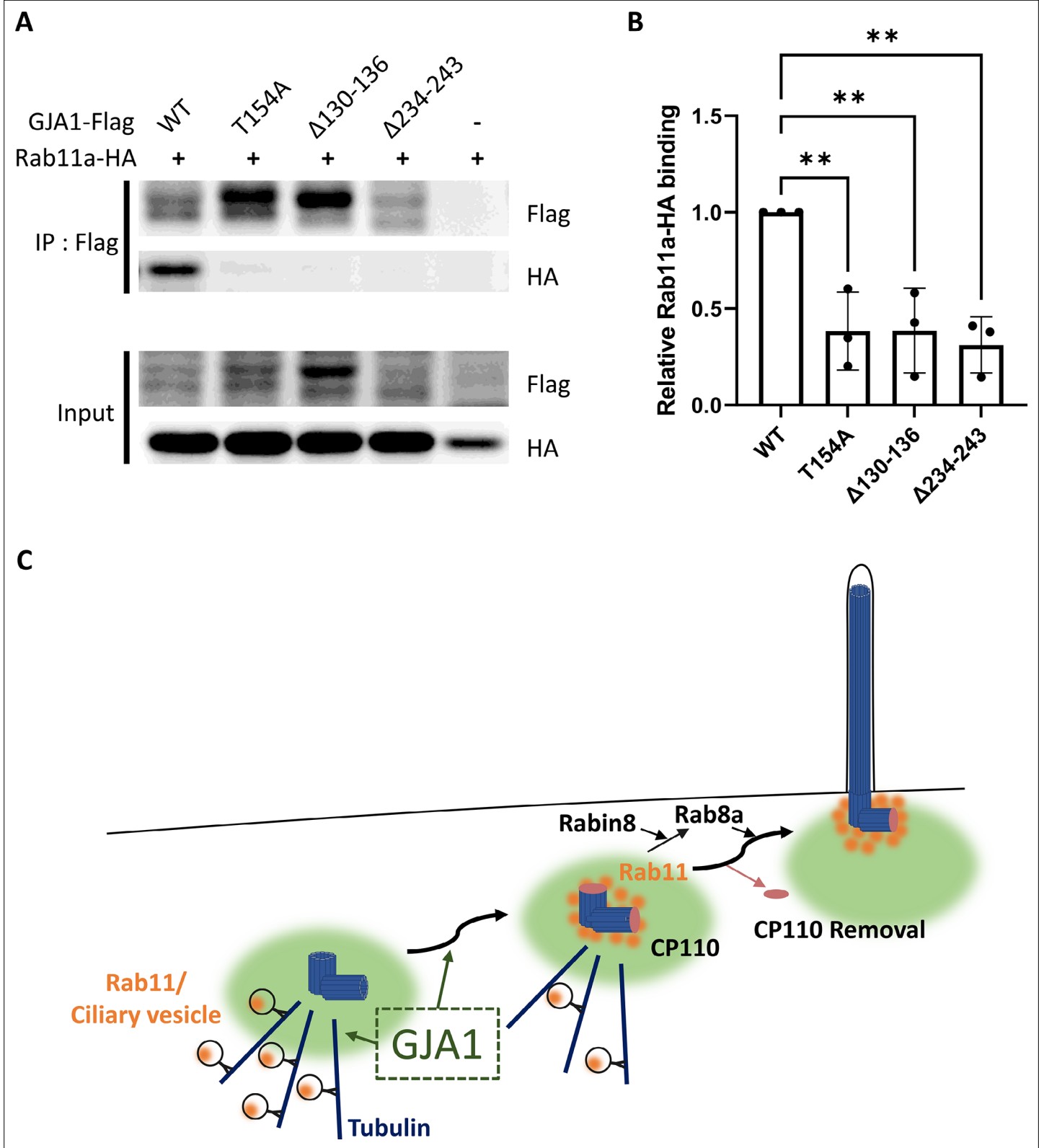

**Figure 8.** Dominant-negative mutant proteins of *Xenopus laevis* GJA1 fail to bind to Rab11. (**A**) HA-tagged *Rab11a* and WT-*GJA1*-Flag or dn*GJA1*-Flag were co-expressed in RPE1 cells, and the lysates were used for immunoprecipitation assays with anti-flag antibody-conjugated beads. The immunoprecipitants were analyzed by immunoblotting with anti-flag or anti-HA antibodies. Rab11a-HA co-immunoprecipitated only with WT-GJA1-Flag. (**B**) Statistical analysis of the relative Rab11a-HA binding affinity to WT-GJA1-Flag or dnGJA1-Flag mutant proteins in panel (**A**). Error bars

*Figure 8 continued on next page*

*Figure 8 continued*

represent the mean ± SD. P values were determined by one-way ANOVA [p**=0.0050 (T154A, Δ130–136), p** = 0.0025 (Δ234–243)]. n=3(WT, T154A, Δ130-136, Δ234-243). Raw values are provided in the *Figure 8—source data 1* file. (**C**) Suggested role of GJA1 during ciliogenesis. GJA1 is necessary for Rab11 positive vesicles trafficking to the vicinity of the basal bodies, thereby affecting CP110 removal to initiate ciliogenesis.

The online version of this article includes the following source data for figure 8:

**Source data 1.** The relative Rab11a-HA binding affinity to WT-GJA1-Flag or dnGJA1-Flag mutant proteins.

**Source data 2.** The uncropped images of western blots.

DMEM/F12 (1:1) media (Gibco). hTERT-RPE-1 cells were authenticated by STR profiling service from Cosmogenetech Inc (Seoul, Republic of Korea). Mycoplasma contamination was monitored with Cell Culture Contamination Detection Kit (Invitrogen, ThermoFisher Scientific) regularly.

## Transfection
For *GJA1* siRNA knockdown experiments, RPE1 cells were transfected with siRNA (Genolution, Seoul, Republic of Korea, sense, 5'-GUUCAAGUACGGUAUUGAAUU-3') using jetPRIME transfection reagent (Polyplus, NY, USA). Exogenous plasmid DNA was transfected using Transporter 5 transfection reagent (Polysciences, PA, USA). Co-transfection of siRNA with plasmid DNA was performed using jetPRIME transfection reagent. All transfections were performed in Opti-MEM medium (Gibco). The final concentration of siRNA was 70 nM. Transfected cells were incubated for 48 hr, including the serum starvation step.

## Ciliation and serum starvation
For primary ciliation of RPE1 cells, the cells were incubated in serum-free DMEM/F12 medium (Gibco) for at least 24 hr.

## Fixation
To analyze primary cilium development, RPE1 cells were fixed in PFA (4% formaldehyde in 1×PBS) fixative solution after ciliation.

## Intracellular $Ca^{2+}$ entry imaging
For intracellular $Ca^{2+}$ imaging, *GJA1* siRNA- or plasmid DNA-transfected RPE1 cells were cultured on fibronectin (Sigma Aldrich, MO, USA)-coated cover glasses. Cells were loaded with 2 μM Fura-2/AM (Invitrogen). Ratiometric $Ca^{2+}$ imaging was performed at 340 and 380 nm in 0 or 2 mM $Ca^{2+}$ Ringer's solution using an IX81 microscope (Olympus, Tokyo, Japan) at room temperature. Images were processed with Metamorph and analyzed with Igor Pro software. $Ca^{2+}$ peaks were calculated by determining the maximum of each trace upon application of 0.5 μM thapsigargin (Santa Cruz, TX, USA) and $Ca^{2+}$, and baseline values before application were subtracted.

## Mouse trachea isolation, vibratome sectioning, and imaging preparation
Mouse tracheal tissues were isolated from wild-type mice. Isolated tracheal tissues were fixed with 4% PFA for 1 hr at room temperature with mild shaking and mounted in 3% low-melting point agarose gel (BioShop, Ontario, Canada) in 1×PBS. Mounted tracheal tissues were sectioned into 100 μm slices with a vibratome (VT1000 S, Leica). Sectioned samples were immunostained and mounted with BA:BB solution (both from Sigma Aldrich) for tissue clearing.

## Cloning and plasmid information
The full-length sequences of the *Xenopus laevis GJA1* and human *GJA1*, *Rab8a*, and *Rab11a* genes were obtained from the Xenbase database or PubMed. Total *Xenopus laevis* RNA was purified from stage 35–36 embryos, and total human RNA was purified from RPE1 cells. Next, we reverse-transcribed cDNA with a random primer mix (NEB, MA, USA) and GoScript Reverse Transcriptase (Promega, WI, USA). We designed the primer pairs with a restriction enzyme site for cloning (*Table 1*). For the rescue experiments, point mutations were generated in the MO- or siRNA-binding sequences of *GJA1* cDNA.

**Table 1.** Primer information.

| Primer Name | Primer Sequences (5′ – 3′) |
| --- | --- |
| SalI-x*GJA1*-Forward | aatgtcgacATGGGTGACTGGAGTGCCTTAGG |
| XbaI-xl*GJA1*-Backward | aattctagaGATCTCTAAATCATCAGGTCGTGGTCT |
| SalI-h*GJA1*-Forward | aatgtcgacATGGGTGACTGGAGCGCCT |
| XbaI-h*GJA1*-Backward | aattctagaGATCTCCAGGTCATCAGGCCG |
| point mut-xl*GJA1*-Forward | CGACATGGGGGATTGGAGCGCATTGGGAAGACTT |
| point mut-xl*GJA1*-Backward | AGTCTTCCCAATGCGCTCCAATCCCCCATGTCGA |
| point mut-h*GJA1*-Forward | TGAGATAAAGAAATTTAAAATATGGAATAGAGGAGCATGTAAGGTGAAA |
| point mut-h*GJA1*-Backward | CTTACCATGCTCCTCTATTCCATATTTAAATTTCTTTATCTCAATCTGC |
| SalI-h*Rab8a*-Forward | aatgtcgacATGGCGAAGACCTACGATTACCTG |
| XbaI-h*Rab8a*-Backward | aattctagaCAGAAGAACACATCGGAAAAAGCTG |
| SalI-h*Rab11a*-Forward | aatgtcgacATGGGCACCCGCGACG |
| XbaI-h*Rab11a*-Backward | aattctagaGATGTTCTGACAGCACTGCACCTT |
| xl*GJA1*-T154A-Forward | AAAGTCAAGATGCGAGGTGGACTGCTTCGCGCCTACATCATCAGCATTTGTTTAAA |
| xl*GJA1*-T154A-Backward | TACTGATTTAAACAAAATGCTGATGATGTAGGCGCGAAGCAGTCCACCTCGCATCTT |
| xl*GJA1*-Δ130–136-Forward | ATGCACCTTAAACAATATGGCCTTGAAGAG |
| xl*GJA1*-Δ130–136-Backward | CTCTTCAAGGCCATATTGTTTAAGGTGCAT |
| xl*GJA1*-Δ234–243-Forward | TTCTATGTCACCTACAAAGACCCATTTCT |
| xl*GJA1*-Δ234–243-Backward | AGAAAATGGGTCTTTGTAGGTGACATAGAA |

We designed the primer pairs to generate a silencing point mutation (*Table 1*). Amplified *Xenopus GJA1* and human *GJA1*, *Rab8a*, and *Rab11a* PCR products were cloned into the CS108 or pcDNA3.0 vectors and fused to Flag- or HA-tag sequences at the C-terminus of the coding sequence. Dominant-negative forms of *GJA1* constructs were generated based on the cloned *Xenopus laevis* wild-type *GJA1*-Flag plasmid with deletion or point mutation primer pairs (*Table 1*). After PCR amplification using the primers, the DpnI enzyme was added, and the PCR product was transformed into DH5α *E. coli*. Wild-type or dominant-negative *GJA1* mRNAs were transcribed from the cloned plasmid using an mMESSAGE mMACHINE SP6 Transcription Kit (Invitrogen).

## Immunofluorescence

Embryos were fixed with MEMFA or Dent's fixative solution overnight at 4°C, and RPE1 cells were fixed with 4% PFA solution for 20 min at room temperature. Fixed embryos or RPE1 cells were incubated in a blocking solution (10% FBS and 2% DMSO in 1×TBS with 0.1% Triton X-100) for 30 min at room temperature to block non-specific binding. Immunostaining was performed with the following antibodies at 1:50–500 dilutions in blocking solution for 1–2 hr at room temperature: anti-DDDD-K (Abcam, Cambridge, UK, ab1162), anti-GFP (ThermoFisher Scientific, A10262), anti-HA (Santa Cruz, sc-7392), anti-RFP (Abcam, ab62341), anti-acetylated tubulin (Sigma Aldrich, T7451), anti-γ-tubulin (Abcam, ab191114), anti-MyHC (DSHB, IA, USA, A4.1025), anti-CP110 (Proteintech, IL, USA, 12780–1-AP), anti-BBS4 (Proteintech, 12766–1-AP), anti-TGN46 (Abcam, ab50595), anti-Arl13b (Proteintech, 17711–1-AP), anti-TTBK2 (Proteintech, 15072–1-AP), anti-IFT20 (Proteintech, 13615–1-AP), anti-IFT88 (Proteintech, 13967–1-AP), anti-Rab8a (Cell Signaling, MA, USA, #6975), anti-Rab11 (Cell Signaling, #5589), and anti-GJA1 (both ThermoFisher Scientific, PA1-25098, 13–8300). Fluorescence labeling was performed using DAPI (1:10,000, ThermoFisher Scientific, D1306), and Alexa Fluor 488-, 555-, and 633-conjugated secondary antibodies (1:300, all ThermoFisher Scientific).

## Immunoblotting

*Xenopus* or human RPE1 immunoblotting samples were prepared with modified lysis buffer (TBS, 1% Triton X-100, and 10% glycerol with protease inhibitor). After removing fat and cellular debris, a sodium dodecyl sulfate (SDS) sample buffer with dithiothreitol (DTT) was added. Samples were loaded on SDS-polyacrylamide gels and transferred to polyvinylidene fluoride membranes (Merck Millipore, MA, USA). Membranes were incubated in blocking solution (TBS, 0.05% Tween-20 with non-fat powdered milk) for 30 min at room temperature to block non-specific binding. Immunoblotting was performed with the following antibodies at 1:2,500–3,000 dilutions for either 1 hr at room temperature or overnight at 4°C: anti-DDDD-K (Abcam, ab1162), anti-α-actin (ThermoFisher Scientific, MA1-744), anti-α-tubulin (Abcam, ab15246), anti-GJA1 (ThermoFisher Scientific, PA1-25098), anti-HA (Santa Cruz, sc-7392), anti-Rab8a (Cell Signaling, #6975), and anti-Rab11 (Cell Signaling, #5589). Secondary labeling was performed using horseradish peroxidase-conjugated anti-mouse or anti-rabbit IgG antibodies (1:3000, both ThermoFisher Scientific, 31430, 31460) for 1 hr at room temperature. Chemiluminescence was performed with SuperSignal West Dura Extended Duration Substrate (ThermoFisher Scientific, 34076), and the membranes were imaged with an iBright imaging system (FL1000, ThermoFisher Scientific).

## Immunoprecipitation and mass spectrometry

For immunoprecipitation of GJA1, we produced anti-GJA1 antibody-conjugated paramagnetic beads. Anti-GJA1 antibody (ThermoFisher Scientific, PA1-25098) or normal rabbit serum (Abclon, Seoul, Republic of Korea) was cross-linked with Protein G Mag Sepharose beads (GE Healthcare, IL, USA), according to the manufacturer's instructions. Human RPE1 lysate was prepared with immunoblotting lysis buffer and incubated with GJA1- or rabbit serum-conjugated beads overnight at 4°C. The beads were washed with lysis buffer and eluted with low-pH elution buffer (0.1 M glycine-HCl, pH 2.9). Next, eluted samples were concentrated using Amicon Ultra (Merck Millipore), and SDS sample buffer and DTT were added. For mass spectrometry, protein samples were separated on a 12% Tris-glycine SDS-polyacrylamide gel. The gel was stained with Coomassie brilliant blue G-250 overnight, after which it was destained with a destaining solution (10% ethanol/2% orthophosphoric acid). Gel pieces were destained in 1:1 methanol:50 mM ammonium bicarbonate, and destained gels were dehydrated with 100% acetonitrile and then incubated in 25 mM DTT in 50 mM ammonium

bicarbonate for 20 min at 56°C. After removing the DTT solution, the samples were alkylated in 25 mM iodoacetamide in 100 mM ammonium bicarbonate solution. The iodoacetamide solution was removed, and then the samples were dehydrated with 100% acetonitrile. Approximately 0.25 µg of mass-spectrometry-grade trypsin protease was added to each sample, along with 100 µL of a 50 mM ammonium bicarbonate/10% acetonitrile solution. Protein samples were then digested overnight at 37°C. After digestion, 100 µL of 50% acetonitrile/0.1% formic acid solution was added to the trypsin solution for 3 min at 56°C. Extraction solution (99.9% acetonitrile/0.1% formic acid; 100 µL) was added to each sample for 3 min at 56°C. Collected peptides were dried and dissolved with 20 µL of 0.1% Trifluoroacetic acid solution and purified with C18 tips. Eluted samples were dried completely in a vacuum concentrator and resuspended with 20 µL of 0.1% formic acid solution. The samples were analyzed with a high-resolution mass spectrometer (Orbitrap Fusion Lumos, ThermoFisher Scientific). Immunoprecipitation analysis of the GJA1 and Rab proteins was performed using anti-GJA1 antibody-conjugated beads, which were described above, and wild-type or dominant-negative GJA1 and Rab11a mutants were detected using anti-Flag M2 magnetic beads (Sigma Aldrich), which captured the GJA1 Flag-tagged proteins, using an equal lysis and immunoprecipitation method. The raw data of GJA1 IP/MS analysis were uploaded to the Dryad (doi:10.5061/dryad.tht76hdxt, *Figure 7—figure supplement 2*).

## Microscopy, image analysis, and statistical analysis

Images were captured by confocal microscopy (LSM780, LSM880, both from Zeiss, Oberkochen, Germany), stereo microscopy (SZX16, Olympus), and SIM (ELYRA S.1, Zeiss). Image analyses were performed using ZEN software for merging, 3D images, and SIM processing. For the randomization of statistical analysis, the images were randomly taken from the samples and individual cells in the wide field of image were analyzed. Quantitative analyses of the average ciliary length and number of multiciliated or RPE1 cells were performed with the ZEN and ImageJ programs. Statistical analysis including error bars with mean ± SD and P values was calculated using a two-tailed t-test, one-way ANOVA, or two-way ANOVA using Prism 9.

## Acknowledgements

The authors thank Dr. M Ko and Dr. SH Park for critical discussions.

This work was supported by the Korea National Research Foundation (2021R1A2B5B02002285) and Institute for Basic Science (IBS-R001-D1).

## Additional information

### Funding

| Funder | Grant reference number | Author |
| --- | --- | --- |
| National Research Foundation of Korea | 2021R1A2B5B02002285 | Tae Joo Park |
| Institute for Basic Science | IBS-R001-D1 | Taejoon Kwon |

The funders had no role in study design, data collection and interpretation, or the decision to submit the work for publication.

### Author contributions

Dong Gil Jang, Conceptualization, Data curation, Software, Formal analysis, Investigation, Methodology, Writing – original draft; Keun Yeong Kwon, Conceptualization, Investigation, Methodology; Yeong Cheon Kweon, Conceptualization, Investigation, Visualization, Methodology; Byung-gyu Kim, Conceptualization, Data curation, Investigation, Methodology; Kyungjae Myung, Conceptualization, Resources, Methodology; Hyun-Shik Lee, Conceptualization, Resources, Supervision; Chan Young Park, Conceptualization, Designing and performing experiments in measuring calcium dynamics; Taejoon Kwon, Conceptualization, Resources, Data curation, Supervision, Methodology, Writing

– original draft; Tae Joo Park, Conceptualization, Resources, Data curation, Supervision, Funding acquisition, Investigation, Writing – original draft, Writing – review and editing

### Author ORCIDs
Taejoon Kwon http://orcid.org/0000-0002-9794-6112
Tae Joo Park http://orcid.org/0000-0003-3176-177X

### Ethics
All animal experiments were performed with appropriate ethical approval from the UNIST Institutional Animal Care and Use Committee (UNISTIACUC-19-22, UNISTIACUC-20-26).

### Decision letter and Author response
Decision letter https://doi.org/10.7554/eLife.81016.sa1
Author response https://doi.org/10.7554/eLife.81016.sa2

---

## Additional files

### Supplementary files
• Supplementary file 1. GJA1 IP-MS results. Ciliogenesis genes (bold) and $Ca^{2+}$ transport genes (*italics*) are indicated.
• MDAR checklist

### Data availability
All data generated or analysed during this study are included in the manuscript, supporting file and source data files.

The following dataset was generated:

| Author(s) | Year | Dataset title | Dataset URL | Database and Identifier |
|---|---|---|---|---|
| Jang D, Park T | 2022 | GJA1 IP/MS dataset | https://doi.org/10.5061/dryad.tht76hdxt | Dryad Digital Repository, 10.5061/dryad.tht76hdxt |

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
