## [Editor Report]

This important paper solidly demonstrates that the gap junction protein GJA1 localizes to motile cilia and is required for the formation of motile cilia on the developing frog epidermis. In addition, GJA1 localizes to the peri-centrosomal region of primary cilia where it is proposed to participate in Rab8-Rab11 delivery of ciliary cargo. These findings point to new functions for gap junction proteins and raise important questions about the role of Gja1 in ciliary assembly and function.

---

## [Decision Letter]

**Decision letter after peer review:**

[Editors’ note: the authors submitted for reconsideration following the decision after peer review. What follows is the decision letter after the first round of review.]

Thank you for submitting your article "GJA1 Depletion Causes Ciliary Defects by Affecting Rab11 Trafficking to the Ciliary Base" for consideration by *eLife*. Your article has been reviewed by 3 peer reviewers, including Gregory J Pazour as the Reviewing Editor and Reviewer #1, and the evaluation has been overseen by Piali Sengupta as the Senior Editor.

The reviewers have discussed the reviews with one another and the Reviewing Editor has drafted this decision to help you prepare a revised submission.

Summary:

In this work, Jang et al. examined the role of the gap junction protein GJA1 in cilia formation. The authors show that GJA1 localizes to motile cilia on the epidermis of frog embryos and that expression of dominant negative constructs or morpholinos against GJA1 reduces the number of cilia present. They further show that loss of GJA1 reduces ciliation in cultured RPE1 cells, which they attribute to Rab11 dysfunction when GJA1 is missing. While all three reviewers found the cilia – gap junction connection exciting and novel, all had concerns about missing controls for critical experiments and none could articulate what function GJA1 might have in ciliogenesis.

Essential revisions:

The localization to frog cilia needs to be better described. The figures need to include panels with only the GJA1 channel so the pattern can be better seen. The images suggest that the label is punctate and possibly at the tip of the cilia. Is this true? What parts of the cilium are labeled? The authors should label the basal bodies (centrin-GFP, chibby-GFP, or anti-γ tubulin antibody) and see if GJA1 co-localizes or not. The figure needs to be labeled as GJA1-Flag to make clear that this is localization of fusion protein rather than endogenous. Does the antibody against mammalian GJA1 cross-react with *Xenopus*? Demonstration of localization of native protein would strengthen the case. Alternatively, the investigators could examine mouse tracheal or ependymal cilia to see if the ciliary localization is conserved in motile cilia.

The *Xenopus* experiments focus on cilia length, which does not seem to be the major parameter affected by disrupting GJA1. A few experiment quantitate cilia loss, but this needs to be done in each of the experiments.

The authors state "Moreover, the MO-mediated knockdown of GJA1 resulted in a smaller number of multi-ciliated cells in *Xenopus* embryonic epithelial tissues (Figure 3B middle panel, 3C)." If the authors are stating there are fewer multiciliated cells, they should show this by MCC cell fate (dnah9, foxj1 in situs) rather than cilia staining – the latter cannot distinguish between loss of cilia and loss of MCC cell fate.

The authors state: "Antisense MO injection effectively reduced the expression level of wild-type GJA1 (Figure 3E, MO + wt-mRNA). In contrast, the translation of mismatch mRNA, which was modified to be mismatched with MO, was not affected as strongly as wild-type GJA1 mRNA by co-injection of GJA1-MO (Figure 3E, MO + mis-mRNA)." This is an unusual way to test MO specificity. First, it is absolutely essential that the authors define whether the MO does or does not have a binding site to the WT-GJA1 that they are injecting for rescue. I must presume that the WT-GJA1-FLAG does indeed have a MO binding site since the expressed protein from this construct is reduced with the MO (Figure 1e). Therefore, the rescue in 1D is meaningless as the injected mRNA is simply binding to the injected MO and does not at all address if the MO is specifically depleting the native protein by binding to the native mRNA. Effectively, there is no real MO controls in this paper. This is unacceptable. The authors much show either phenocopy using a second non-overlapping MO, F0 CRISPR, rescue with an mRNA that does not bind the MO and ideally a combination of all of the above.

In RPE1 cells, GJA1 does not localize to cilia but rather localizes in a diffuse area around the base of cilia. The authors say this is pericentriolar material. It is hard to tell from the images, but it could just as likely be the Golgi complex. Co stain with peri-centriolar and Golgi markers are needed. Why were the colors changed between 5A and B? Higher mag images of 5B are needed to determine if the "pericentriolar" label is missing in the knockdown. Many of the cells in the center panel still show this label.

Multiple localization proteins: Studying proteins that have multiple localizations (in this case at the gap junction and possibly at the bases of cilia) is a challenge. The authors suggest that the function of GJ1A is due to its role at the bases of cilia. However, to be able to show that this is in fact the case requires constructs that either localize only to one location (by either adding sub cellular localization sites or deleting localization sites) and showing rescue of a loss of function phenotype. OR they need to test mutant constructs that abrogate subcellular specific function that nonetheless rescue the function of interest in the context of gene depletion. For example, mutant constructs that interfere with gap junction function but RESCUE the loss of cilia in GJA1 depleted cells. Unfortunately, for the author's argument, they have exactly the opposite. Injection of dnGJA1 constructs that interfere with gap junction function also eliminates cilia suggesting that it is in fact the gap junction role that is critical for cilia not any cilia localization. The authors claim that they do not see calcium defects and their mass spec does identify genes that are associated with cilia function. However, this may be true but unrelated. Of course, it is possible that the domains of GJA1 that are relevant for gap junction function are also critical for a function at the ciliary base – however, the simplest explanation is that the GJA1 dominant negative constructs point towards a role at the gap junction. Many experiments could be done to address this. Do the dnGJA1 proteins bind to the cilia proteins identified? Do the dnGJA1 proteins also localize to the cilia bases?

The model and the discussion need to more clearly state what the authors think GJA1 is doing to regulate ciliogenesis. The model in S7 suggests that it might be doing something with microtubules but this was not developed in the paper.

---

## [Author Response]

[Editors’ note: the authors resubmitted a revised version of the paper for consideration. What follows is the authors’ response to the first round of review.]

Essential revisions:The localization to frog cilia needs to be better described. The figures need to include panels with only the GJA1 channel so the pattern can be better seen. The images suggest that the label is punctate and possibly at the tip of the cilia. Is this true? What parts of the cilium are labeled? The authors should label the basal bodies (centrin-GFP, chibby-GFP, or anti-γ tubulin antibody) and see if GJA1 co-localizes or not. The figure needs to be labeled as GJA1-Flag to make clear that this is localization of fusion protein rather than endogenous.

1. We added panels showing the GJA1 signal to each figure to improve readability.

2. We observed that GJA1 was localized at ciliary axonemes but not exclusively at the tip of cilia. This finding was cross-checked using both Flag- and hemagglutinin (HA)-tagged GJA1 proteins (Figure 1B-D).

Additionally, we observed differential localization of GJA1 dominant-negative mutants (dnGJA1) in multiciliated cells. The Δ234–243 GJA1 mutant failed to exhibit typical GJA1 localization to ciliary axonemes, and the Δ130–136 mutant displayed reduced GJA1 localization at ciliary axonemes (Figure 2—figure supplement 1D), which indicates that GJA1 localization in the cilia may be regulated. In addition, we showed that endogenous GJA1 is also localized at ciliary axonemes in the mouse tracheal multiciliated cells (Figure 1—figure supplement 1B).

3. We added data showing GJA1 localization with the basal body markers, centrin-GFP and an anti-γ-tubulin antibody (Figure 1B–D, Figure 1—figure supplement 1A). The data showed that GJA1 is localized near the centrin signal but does not overlap with centrin.

4. We corrected the corresponding sentences and figure legends and described the use of GJA1-Flag to clearly explain the exogenous GJA1 fusion protein.

Does the antibody against mammalian GJA1 cross-react with *Xenopus*? Demonstration of localization of native protein would strengthen the case. Alternatively, the investigators could examine mouse tracheal or ependymal cilia to see if the ciliary localization is conserved in motile cilia.

Unfortunately, the two GJA1 antibodies from mouse and rabbit hosts did not cross-react with *Xenopus laevis* GJA1 in the immunofluorescence experiments. We examined GJA1 localization in the mouse tracheal mucociliary epithelium and observed that GJA1 localizes to the ciliary axonemes (Figure 1—figure supplement 1B).

The *Xenopus* experiments focus on cilia length, which does not seem to be the major parameter affected by disrupting GJA1. A few experiment quantitate cilia loss, but this needs to be done in each of the experiments.

The authors state "Moreover, the MO-mediated knockdown of GJA1 resulted in a smaller number of multi-ciliated cells in *Xenopus* embryonic epithelial tissues (Figure 3B middle panel, 3C)." If the authors are stating there are fewer multiciliated cells, they should show this by MCC cell fate (dnah9, foxj1 in situs) rather than cilia staining – the latter cannot distinguish between loss of cilia and loss of MCC cell fate.

As the reviewer suggested, we performed DNAH9 in situ hybridization and observed that the number of DNAH9-positive cells was not significantly different in GJA1 morphant embryos (Figure 3—figure supplement 1A, B). These data indicate that GJA1 may not directly affect cell fate determination but rather that GJA1 is necessary for cilia formation.

We revised the indicated sentence and discussed these results in the “Results” section.

The authors state: "Antisense MO injection effectively reduced the expression level of wild-type GJA1 (Figure 3E, MO + wt-mRNA). In contrast, the translation of mismatch mRNA, which was modified to be mismatched with MO, was not affected as strongly as wild-type GJA1 mRNA by co-injection of GJA1-MO (Figure 3E, MO + mis-mRNA)." This is an unusual way to test MO specificity. First, it is absolutely essential that the authors define whether the MO does or does not have a binding site to the WT-GJA1 that they are injecting for rescue. I must presume that the WT-GJA1-FLAG does indeed have a MO binding site since the expressed protein from this construct is reduced with the MO (Figure 1e). Therefore, the rescue in 1D is meaningless as the injected mRNA is simply binding to the injected MO and does not at all address if the MO is specifically depleting the native protein by binding to the native mRNA. Effectively, there is no real MO controls in this paper. This is unacceptable. The authors much show either phenocopy using a second non-overlapping MO, F0 CRISPR, rescue with an mRNA that does not bind the MO and ideally a combination of all of the above.

We understand the reviewer’s concern, and we apologize for not adequately explaining the detailed procedure of the rescue experiments.

In the “Materials and methods” section of the previous manuscript, we described that “For the rescue experiments, point mutations were generated in the MO- or siRNA-binding sequences of GJA1 cDNA”. Additionally, we included the primer sequences used to generate point mutations, which are mismatched to the GJA1-MO or siRNA sequences as shown in “Table 1. Primer information”. All rescue experiments in *Xenopus* and RPE1 cells were performed with point mutant mRNA or cDNA which are not targetable by the GJA1-MO or siRNA.

In the revised manuscript, we explained that we used MO or siRNA non-targetable GJA1 cDNA for the rescue experiments. In addition, we included F0 mutagenesis data generated by CRISPR/Cas9 targeting of GJA1 (Figure 3—figure supplement 2). Overall, we have confirmed the loss-of-function of GJA1 using three different approaches, MO-mediated depletion, siRNA-mediated depletion, and CRISPR/cas9-mediated F0 deletion. Additionally, we showed rescue of GJA1 loss-of-function phenotypes in *Xenopus* and RPE1 cells. Altogether, we believe that defects in cilia formation comprise a specific phenotype of GJA1 depletion.

In RPE1 cells, GJA1 does not localize to cilia but rather localizes in a diffuse area around the base of cilia. The authors say this is pericentriolar material. It is hard to tell from the images, but it could just as likely be the Golgi complex. Co stain with peri-centriolar and Golgi markers are needed. Why were the colors changed between 5A and B? Higher mag images of 5B are needed to determine if the "pericentriolar" label is missing in the knockdown. Many of the cells in the center panel still show this label.

1. In response to the reviewer’s comments, we carefully re-examined GJA1 localization with the PCM and Golgi markers in RPE1 cells. As the reviewer noted, we observed that the GJA1 signal is partially co-localized with the Golgi marker. However, the GJA1 signal is also broadly dispersed around the pericentriolar regions, although not specifically at the PCM, which is marked by BBS4 (BBS4 is known to co-localize with PCM-1; Nachoory et al., Cell. 2007) (Figure 5C, Figure 5—figure supplement 2A).

2. In this revised manuscript, we used the same color in Figure 5A and C. And added the GJA1 signal panels to Figure 5A and C to better illustrate the GJA1 pericentriolar label. Although we observed a significant reduction in the GJA1 signal in most GJA1 siRNA-treated cells, GJA1 signals were still detectable. We believe that these signals were detectable because of the incomplete depletion of GJA1 by siRNA treatment. Nonetheless, our data clearly show an overall and significant reduction of GJA1 expression in siRNA-treated cells.

Multiple localization proteins: Studying proteins that have multiple localizations (in this case at the gap junction and possibly at the bases of cilia) is a challenge. The authors suggest that the function of GJ1A is due to its role at the bases of cilia. However, to be able to show that this is in fact the case requires constructs that either localize only to one location (by either adding sub cellular localization sites or deleting localization sites) and showing rescue of a loss of function phenotype. OR they need to test mutant constructs that abrogate subcellular specific function that nonetheless rescue the function of interest in the context of gene depletion. For example, mutant constructs that interfere with gap junction function but RESCUE the loss of cilia in GJA1 depleted cells. Unfortunately, for the author's argument, they have exactly the opposite. Injection of dnGJA1 constructs that interfere with gap junction function also eliminates cilia suggesting that it is in fact the gap junction role that is critical for cilia not any cilia localization. The authors claim that they do not see calcium defects and their mass spec does identify genes that are associated with cilia function. However, this may be true but unrelated. Of course, it is possible that the domains of GJA1 that are relevant for gap junction function are also critical for a function at the ciliary base – however, the simplest explanation is that the GJA1 dominant negative constructs point towards a role at the gap junction. Many experiments could be done to address this. Do the dnGJA1 proteins bind to the cilia proteins identified? Do the dnGJA1 proteins also localize to the cilia bases?

We appreciate the reviewer’s critical suggestion for further study to strengthen our main idea and experiments. Indeed, our data showed that all three dnGJA1 mutants failed to recover the ciliary defects in GJA1 morphant embryos. Additionally, overexpression of each GJA1 mutant interfered with normal ciliogenesis, although there seemed to be some differences in the severity of ciliary defects (the Δ234–243 mutant caused relatively severe ciliary defects, although it would be challenging to quantitatively analyze the phenotypes) (Figure 2C, D).

In this revised manuscript, we included several lines of evidence that suggest how GJA1 mutants interfere with wild-type GJA1(wtGJA1) function on ciliogenesis.

First, we show differential localization of the dnGJA1 mutants (Figure 2—figure supplement 1D), and also show that overexpression of dnGJA1 mutants displaced the ciliary localization of wtGJA1 (Figure 2C, D).

In addition, we performed co-immunoprecipitation experiments with GJA1 mutants and showed that these mutants failed to interact with Rab11a (Figure 8A, B). These data may explain why all the GJA1 mutants affected cilia formation and were not able to rescue ciliary defects caused by the loss-of-function of GJA1.

Both Rab8 and Rab11 have been reported to be involved in ciliogenesis [25, 26, 37, 38]. Previous research has also shown that Rab8 interacts with CP110 and BBS4 [34, 35, 37], and Rab11 is enriched at the base of the primary cilium and recruits Rab8 to the basal body by interacting with Rabin8 [26, 38].

Additionally, during basal body maturation, Rab11-positive membrane vesicles are transported to the distal appendages of basal bodies and form large ciliary vesicles. This distal appendage vesicle assembly that forms ciliary vesicles is a prerequisite for CP110 removal and promotes the uncapping of the mother centrioles, which is followed by the recruitment of intraflagellar transport proteins and transition zone proteins for ciliogenesis (Irma Sánchez et al. Nat Cell Biol, 2016; Quanlong Lu et a., Nat Cell Biol. 2015).

We regret that we were not able to clearly identify which domain of GJA1 may be responsible for the interaction with Rab11/Rab8 and critical for ciliogenesis. However, our data suggest that GJA1 regulates pericentriolar vesicle trafficking, which affects Rab11-positive ciliary vesicle accumulation around the pericentriolar regions. Thus, CP110 removal is attenuated, and consequently, ciliogenesis is inhibited. We believe that this finding is an important discovery and will make a strong impact on research in a broad spectrum of biomedical science.

The model and the discussion need to more clearly state what the authors think GJA1 is doing to regulate ciliogenesis. The model in S7 suggests that it might be doing something with microtubules but this was not developed in the paper.

In response to the reviewer’s comments, we modified the figure and discussed the possible mechanism by which GJA1 may be involved in ciliogenesis.

Our data show that the Δ234–243 mutant, which is formed by deletion of the microtubule-binding domain, completely failed to localize at ciliary axonemes (Figure 2C, D) and the Δ234–243 mutant caused severer ciliary defects. We suggest that microtubules are essential for the proper localization and function of GJA1 in ciliogenesis.